# Towards Generalized Certified Robustness with Multi-Norm Training

**Enyi Jiang, David S. Cheung, Gagandeep Singh**
**University of Illinois at Urbana-Champaign**
`{enyij2,davidsc2,ggnds}@illinois.edu`

Reviewed on OpenReview: `https://openreview.net/forum?id=U5U7pazr6X`

## Abstract

Existing certified training methods can only train models to be robust against a certain perturbation type (e.g. $l_\infty$ or $l_2$). However, an $l_\infty$ certifiably robust model may not be certifiably robust against $l_2$ perturbation (and vice versa) and also has low robustness against other perturbations (e.g. geometric and patch transformation). By constructing a theoretical framework to analyze and mitigate the tradeoff, we propose the first multi-norm certified training framework **CURE**, consisting of several multi-norm certified training methods, to attain better *union robustness* when training from scratch or fine-tuning a pre-trained certified model. Inspired by our theoretical findings, we devise bound alignment and connect natural training with certified training for better union robustness. Compared with SOTA-certified training, **CURE** improves union robustness to 32.0% on MNIST, 25.8% on CIFAR-10, and 10.6% on TinyImagenet across different epsilon values. It leads to better generalization on a diverse set of challenging unseen geometric and patch perturbations to 6.8% and 16.0% on CIFAR-10. Overall, our contributions pave a path towards *generalized certified robustness*.

## 1 Introduction

While deep neural networks (DNNs) are widely deployed in various vision applications, they remain vulnerable to adversarial attacks (Goodfellow et al., 2014; Kurakin et al., 2018). Many empirical defenses (Madry et al., 2017; Zhang et al., 2019a; Wang et al., 2023) against adversarial attacks have been proposed, however, they do not provide provable guarantees and remain vulnerable to stronger attacks. Hence, it is important to train DNNs to be *formally* robust against adversarial perturbations. Various deterministic certified training methods for specific perturbations (Mirman et al., 2018; Gowal et al., 2018; Zhang et al., 2019b; Balunović & Vechev, 2020; Shi et al., 2021; Müller et al., 2022; Yang et al., 2022; Hu et al., 2023; 2024; Mao et al., 2024b)(e.g., $l_\infty$, $l_2$, and geometric transformations) have been proposed. However, those defenses are mostly limited to a specific perturbation and cannot easily be generalized to other perturbation types (Yang et al., 2022; Chiang et al., 2020). Multi-norm attacks that examine models' robustness against $l_p$ norms simultaneously have arisen in real-world settings such as cybersecurity Zhang et al. (2024), video recognition Lo & Patel (2021), and social media filtering Dai et al. (2024): it is essential for building models that are robust across diverse $l_p$ norms, to generalize better against other non-$l_p$ perturbations (Jiang & Singh, 2024).

In this work, we propose the first multi-norm **C**ertified training for **U**nion **R**obustn**E**ss (**CURE**) framework, consisting of several multi-norm certified training methods. Inspired by SABR (Müller et al., 2022), we use a deterministic $l_2$ defense that first finds the $l_2$ adversarial examples in a slightly truncated $l_2$ region and then propagates the smaller $l_\infty$ box using the IBP loss (Gowal et al., 2018). In Figure 1a, we show that an $l_\infty$ certified robust model may lack $l_2$ certified robustness and vice versa: $l_\infty$ model only has 6.0% $l_2$ robustness and $l_2$ model has 0% $l_\infty$ robustness, which reveals the **robustness tradeoff** among different $l_p$ perturbations. Therefore, we first construct a theoretical framework for binary classification to analyze the tradeoff, from which we propose several methods based on multi-norm empirical defenses with different loss

formulations (Tramer & Boneh, 2019; Madaan et al., 2021; Croce & Hein, 2022; Jiang & Singh, 2024). Our proposed methods successfully improve union and generalized certified robustness, shown in Table 1, Figure 4, and Table 3a.

However, the aforementioned methods achieve sub-optimal union robustness since they do not exploit the in-depth connections between certified training for different $l_p$ perturbations as well as natural training. Thus, we propose the following improvements. **(1) Bound alignment:** Inspired by the upper bound of theoretical analysis (Theorem 4.2) and previous work on adversarial robustness (Jiang & Singh, 2024), we propose a new *bound alignment* method to mitigate the $l_q - l_r$ tradeoff better. We regularize the distributions of output bound differences, computed with IBP, for $l_q, l_r$ perturbations on the correctly certified subset $\gamma$, as shown in Figure 1b. In this way, we encourage the model to *emphasize optimizing* the samples that can potentially become certifiably robust against multi-norm perturbations. To achieve this, we use a KL loss to encourage the distributions of the $l_q, l_r$ output bound differences on subset $\gamma$ to be close to each other for better union accuracy. **(2) Gradient Projection:** We find that there exist some useful components in natural training that can be extracted and leveraged to improve certified robustness (Jiang & Singh, 2024). To achieve this, we find and incorporate the layer-wise useful natural training components by comparing the similarity of the certified and natural training model updates. **(3) Quick fine-tuning:** Fine-tuning an $l_p$-robust model using bound alignment quickly achieves superior multi-norm certified robustness. By addressing the $l_q - l_r$ tradeoff, bound alignment preserves more $l_q$ robustness when fine-tuning with $l_r$ perturbations, focusing on correctly certified samples. This technique enables efficient multi-norm robustness using pre-trained models with single $l_p$ robustness. Figure 1a shows that both scratch training (CURE-Scratch) and fine-tuning (CURE-Finetune) significantly enhance union robustness over single-norm training. **(4) Generalized robustness:** As a perhaps surprising side effect, improving union-certified robustness leads to stronger *generalized certified robustness* by generalizing better to other geometric and patch transformations (Section 5.1), confirming that $l_p$ robustness is the bedrock for non-$l_p$ robustness (that non-$\ell_p$ perturbations may be modeled through $\ell_p$-bounded formulations) (Mangal et al., 2023).

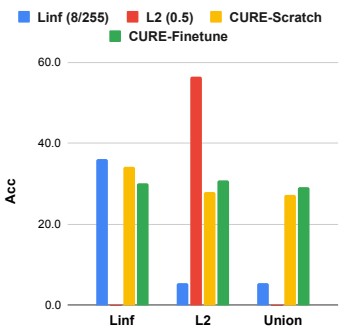

(a) $l_\infty - l_2$ tradeoff on CIFAR10 with $\epsilon_\infty = \frac{8}{255}, \epsilon_2 = 0.5$. The X-axis represents $l_\infty$, $l_2$, and union robustness; different colors refer to different training methods.

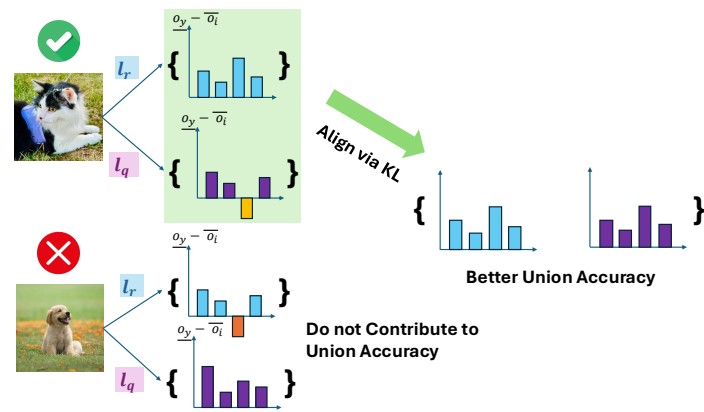

(b) Bound alignment during training.

Figure 1: (a) $l_\infty - l_2$ tradeoff: an $l_\infty$ certified robust model may lack $l_2$ certified robustness and vice versa. **CURE-Scratch** (yellow) and **CURE-Finetune** (green) improve union robustness significantly. (b) We align the output bound differences for $l_q, l_r$ perturbations on the correctly certified $l_q$ subset $\gamma$ to mitigate $l_q - l_r$ tradeoff for better union robustness.

**Main Contributions:**

- We design a theoretical framework to analyze the multi-norm certified robustness tradeoff. Based on this, we propose three training methods, CURE-Joint, CURE-Max, and CURE-Random with different loss formulations for better union and generalized certified robustness.

- Inspired by our theoretical findings, we introduce techniques including bound alignment, connecting natural training with certified training, and certified fine-tuning for better union robustness. CURE-Scratch and CURE-Finetune further facilitate our multi-norm certified training procedure and advance multi-norm robustness.

- Compared with a SOTA certified training method (Müller et al., 2022), **CURE** improves union robustness up to 32.0% on MNIST, 25.8% on CIFAR-10, and 10.6% on TinyImagenet. It improves robustness against unseen geometric and patch perturbations up to $0.6\%, 8.5\%$ on MNIST and $6.8\%, 16\%$ on CIFAR-10.

## 2  Background

In this section, we provide the necessary background of neural network verification and certified training. Given samples $\{(x_i, y_i)\}_{i=0}^{N}$ from a data distribution $\mathcal{D}$, the input comprises images $x \in \mathbb{R}^d$ with labels $y \in \mathbb{R}^k$. The goal is to train a classifier $f$, parameterized by $\theta$, to minimize a loss function $\mathcal{L} : \mathbb{R}^k \times \mathbb{R}^k \to \mathbb{R}$ over $\mathcal{D}$.

### 2.1  Neural network verification

Neural network verification formally proves a network's robustness, with the provably robust samples defining the *certified accuracy*. Interval Bound Propagation (IBP) (Gowal et al., 2018; Mirman et al., 2018) is a simple yet effective method for verification. It over-approximates the input region $B_p(x, \epsilon_p), p \in \{2, \infty\}$, propagates it layer by layer through the network $f = L_j \circ \sigma \circ L_{j-2} \circ \ldots \circ L_1$ (with linear layers $L_i$ and ReLU activations $\sigma$), and verifies whether the reachable outputs classify correctly. Robustness is certified if the lower bound of the correct class exceeds the upper bounds of all others ($\forall i \neq y, \overline{o}_i - \underline{o}_y < 0$) (for more details, see Gowal et al. (2018)).

### 2.2  Training for robustness

A classifier is adversarially robust on an $l_p$-norm ball $B_p(x, \epsilon_p) = \{x' \in \mathbb{R}^d : \|x' - x\|_p \leq \epsilon_p\}$ if it correctly classifies all points within this region, i.e., $\arg\max f(x') = y$ for all $x' \in B_p(x, \epsilon_p)$. Training for robustness is framed as a min-max optimization problem, defined for an $l_p$ attack as:

$$\min_{\theta} \mathbb{E}_{(x,y) \sim \mathcal{D}} \left[ \max_{x' \in B_p(x, \epsilon_p)} \mathcal{L}(f(x'), y) \right] \tag{1}$$

The inner maximization problem is often approximated through adversarial training (Madry et al., 2017) or certified training (Gowal et al., 2018; Müller et al., 2022). However, such methods are typically tailored to specific $p$ values, leaving networks vulnerable to other perturbations. To address this, prior work has only trained networks to be *adversarially* robust against multiple perturbations ($l_1, l_2, l_\infty$). Our focus is on training networks to be *certifiably* robust to multiple $l_p$ perturbations.

### 2.3  Certified training

There are two main categories of methods to train certifiably robust models: unsound and sound methods. Sound methods optimize a rigorously defined upper bound of the inner maximization problem, ensuring provable robustness guarantees. In contrast, unsound methods give up this guarantee to have a more precise approximation. IBP, a sound method, optimizes the following loss function based on logit differences:

$$\mathcal{L}_{\text{IBP}}(x, y, \epsilon_\infty) = \ln(1 + \sum_{i \neq y} e^{\overline{o}_i - \underline{o}_y}) \tag{2}$$

Also, state-of-the-art certified training methods SABR (Müller et al., 2022), TAPs (Mao et al., 2024b), and CC/EXP/MTL-IBP (Palma et al., 2024) relax the robustness guarantee within the specification loss, but in practice, result in better standard and certified accuracy. Given a small box size $\tau_\infty$, SABR finds an

adversarial example $x' \in B_\infty(x, \epsilon_\infty - \tau_\infty)$ and propagates a small box region $B_\infty(x', \tau_\infty)$ across all layers using IBP loss, expressed as:

$$\mathcal{L}_{l_\infty}(x, y, \epsilon_\infty, \tau_\infty) = \max_{x' \in B_\infty(x, \epsilon_\infty - \tau_\infty)} \mathcal{L}_{\text{IBP}}(x', y, \tau_\infty) \tag{3}$$

### 2.4 Evaluation metrics

**Union certified accuracy (UCA).** We focus on the union threat model $\Delta = B_1(x, \epsilon_1) \cup B_2(x, \epsilon_2) \cup B_\infty(x, \epsilon_\infty)$ which requires the DNN to be *certifiably* robust within the $l_1$, $l_2$ and $l_\infty$ adversarial regions simultaneously. Union accuracy is then defined as the robustness against $\Delta_{(i)}$ for each $x_i$ sampled from $\mathcal{D}$. In this paper, similar to the prior works (Croce & Hein, 2022), we use union accuracy as the main metric to evaluate the multi-norm *certified* robustness.

$$\textbf{UCA} = \mathbb{E}_{x_i \sim \mathcal{D}} \left[ \mathbf{1}\{\forall x' \in \Delta \text{ with bounds } \overline{o}_i, \underline{o}_i, \forall i \neq y_i, \overline{o}_i - \underline{o}_{y_i} < 0\} \right],$$

where $y_i$ is the true label for sample $x_i$, and $\mathbf{1}\{\cdot\}$ is the indicator function.

**Generalized certified robustness (GCR).** We measure the generalization ability of multi-norm certified training to other perturbation types, including rotation, translation, scaling, shearing, contrast, and brightness change of geometric transformations (Balunovic et al., 2019; Yang et al., 2022), as well as patch attacks (Chiang et al., 2020). Having perturbation sets $T_j(x)$ representing each transformation or attack type $j$, we define:

$$\textbf{GCR} = \mathbb{E}_{x_i \sim \mathcal{D}} \left[ \frac{1}{J} \sum j = 1^J \mathbf{1}\{\forall x' \in T_j(x_i) \text{ with bounds } \overline{o}_i, \underline{o}_i, \forall i \neq y_i, \overline{o}_i - \underline{o}_{y_i} < 0\} \right],$$

where $J$ is the total number of considered perturbation types.

## 3 Related Work

**Neural network verification.** We rely on deterministic verification techniques to evaluate robustness under multiple norms. Although exact verification is NP-complete and infeasible for large models (Katz et al., 2017), scalable relaxations such as abstract interpretation (Singh et al., 2019) and convex optimization approaches (Wang et al., 2021) make it possible to obtain sound, though sometimes conservative, certificates. These methods are widely used in certified training because they strike a balance between tractability and rigor, enabling provable guarantees at scale. Our analysis of multi-norm certified training builds on this foundation, leveraging deterministic verification to provide stronger and more general robustness guarantees.

**Certified training.** For $l_\infty$ certified training, a widely-used method IBP (Mirman et al., 2018; Gowal et al., 2018) minimizes a sound over-approximation of the worst-case loss, calculated using the Box relaxation method. Wong et al. (2018) applies DeepZ (Singh et al., 2018) relaxations, estimating using Cauchy random projections. CROWN-IBP (Zhang et al., 2019b) integrates efficient Box propagation with precise linear relaxation-based bounds during the backward pass to estimate the worst-case loss. Balunović & Vechev (2020) consists of a verifier that aims to certify the network using convex relaxation and an adversary that tries to find inputs causing verification to fail. Shi et al. (2021) proposes a new weight initialization method for IBP, adds Batch Normalization (BN) to each layer and designs regularization with a short warmup schedule. Besides this, SABR (Müller et al., 2022), TAPS (Mao et al., 2024b), and others De Palma et al. (2022); Mao et al. (2023); Balauca et al. (2024); Mao et al. (2024a) are unsound improvements over IBP by connecting IBP to adversarial attacks and adversarial training. For $l_2$ deterministic certified training, recent works (Leino et al., 2021; Xu et al., 2022; Hu et al., 2023; 2024) are based on Lipschitz-based certification methods. They design specialized architectures under a particular $l_p$ norm, which do not naturally extend to robustness under the diverse settings considered in our work. To the best of our knowledge, **CURE** is the first deterministic framework for multi-norm certified robustness, compatible with diverse model architectures. In comparison to previous works, **CURE** is a more general deterministic framework for multi-norm certified robustness.

**Robustness against multiple perturbations.** Adversarial Training (AT) usually employs gradient descent to discover adversarial examples and incorporates them into training for enhanced adversarial robustness (Tramèr et al., 2017; Madry et al., 2017). Numerous works focus on improving robustness (Zhang et al., 2019a; Carmon et al., 2019; Raghunathan et al., 2020; Wang et al., 2020; Wu et al., 2020; Gowal et al., 2020; Zhang et al., 2021; Debenedetti & Troncoso—EPFL, 2022; Peng et al., 2023; Wang et al., 2023) against a *single* perturbation type while remaining vulnerable to other types. Tramer & Boneh (2019); Kang et al. (2019) observe that robustness against $l_p$ attacks does not necessarily transfer to other $l_q$ attacks ($q \neq p$). Previous studies (Tramer & Boneh, 2019; Maini et al., 2020; Madaan et al., 2021; Croce & Hein, 2022; Jiang & Singh, 2024) modified Adversarial Training (AT) to enhance robustness against multiple $l_p$ attacks, employing average-case (Tramer & Boneh, 2019), worst-case (Tramer & Boneh, 2019; Maini et al., 2020; Jiang & Singh, 2024), and random-sampled (Madaan et al., 2021; Croce & Hein, 2022) defenses. There are also works (Nandy et al., 2020; Liu et al., 2020; Xu et al., 2021; Xiao et al., 2022; Maini et al., 2022) that use preprocessing, ensemble methods, mixture of experts, and stability analysis to solve this problem. For multi-norm certified robustness, Nandi et al. (2023) study the certified multi-norm robustness with probabilistic guarantees. They apply randomized smoothing, which is expensive to compute in nature, making it impractical for real-world applications. Our work in contrast to these works, proposes the first *deterministic* certified multi-norm training for better multi-norm and generalized certified robustness.

## 4    CURE: multi-norm Certified training for Union RobustnEss

This section presents our multi-norm certified training (CT) framework **CURE**. We introduce our framework for binary classification to analyze the trade-off between certified $l_p$ and $l_q$ perturbations, inspired by previous work on the accuracy-robustness trade-off (Zhang et al., 2019a). However, we note our algorithms presented in this work are all multi-class, and the binary classification framework can be easily extended to the multi-class case Zhang et al. (2019a). Based on the theoretical analysis (Eq. 4), we propose three methods for multi-norm CT against $l_2, l_\infty$ perturbations using different loss formulations, serving as the base instantiations of our framework. We design new techniques to improve union-certified accuracy inspired by our theoretical findings.

**Notations.** We denote the sample instance as $x \in \mathcal{X}$, with the label $y \in \{-1, +1\}$ (binary classification), where $\mathcal{X} \subseteq \mathbb{R}^d$ is the instance space. $D = \{(x_i, y_i)\}_{i=1}^n$ is the dataset, where $X = \{x_1, \ldots, x_n\} \subseteq \mathcal{X}$ is the set of instances and $Y = \{y_1, \ldots, y_n\} \subseteq \{-1, +1\}$ is the set of corresponding labels. Let $f : \mathcal{X} \to \mathbb{R}$ map instances to output values $\in \{-1, +1\}$, which can be parameterized (e.g., by neural networks). We use $\mathbf{1}\{\text{event}\}$, the 0-1 loss, as an indicator function that is 1 if an event happens and 0 otherwise. For any function $\psi(\boldsymbol{u})$, we use $\psi^{-1}$ to denote the inverse function. $\phi(\cdot)$ is used to denote the surrogate for the 0-1 loss function.

**Robust, alignment and union error.** To characterize the robustness of a score function $f : \mathcal{X} \to \mathbb{R}$, similar to Schmidt et al. (2018); Cullina et al. (2018); Bubeck et al. (2019), we define *robust error* under the threat model of $\epsilon_q$ perturbation: $\mathcal{R}_q(f) := \mathbb{E}_{(x,y)\sim\mathcal{D}}\mathbf{1}\{\exists x'_q \in B_q(x, \epsilon_q) \text{ s.t. } f(x'_q)y \leq 0\}$. We define $\mathcal{R}_r(f)$ similarly to $\mathcal{R}_q(f)$ for $\epsilon_r$ perturbation, and without loss of generality, assume $\mathcal{R}_r(f) \geq \mathcal{R}_q(f)$. Then, we introduce *alignment error* as the risk calculated by $x \in X$ that are robust against $l_r$ attack but not robust against $l_q$ attack: $\mathcal{R}_{\text{align}}(f) := \mathbb{E}_{(x,y)\sim\mathcal{D}}\mathbf{1}\{\exists x'_r \in B_r(x, \epsilon_r), x'_q \in B_q(x, \epsilon_q), \text{ s.t. } f(x'_r)y > 0 \text{ and } f(x'_q)y \leq 0\}$. The *union error* is the risk calculated by $x \in X$ that are either not robust against $l_q$ or $l_r$ attack. We have the following relationship of $\mathcal{R}_{\text{union}}(f)$:

$$\mathcal{R}_{\text{union}}(f) = \mathcal{R}_r(f) + \mathcal{R}_{\text{align}}(f). \tag{4}$$

**Trade-off between $l_q, l_r$ perturbations.** Our study is motivated by the trade-off between $l_q$ and $l_r$ robust errors (Figure 1a), as well as the accuracy-robustness tradeoff analysis (Zhang et al., 2019a). To illustrate, we provide two extreme cases in Figure 2. We define $S_r = \{x | \exists x'_r \in B_r(x, \epsilon_r) \text{ s.t. } f(x'_r)y \leq 0, (x, y) \in D\}$ (define $S_q$ similarly). As shown in Table 3, we have (a) the *lowest* union accuracy of 0: when all instances in $X$ can be successfully attacked by $l_q$ or $l_r$ norms yet no single instance can be attacked in both $l_q$ and $l_r$, we have a union error of 1; (b) the *highest* union accuracy of $1 - \mathcal{R}_r$: when the instances that are not robust against $l_r$ attack includes all instances that are not robust against $l_q$ attack, we have a union error $\mathcal{R}_r$. A larger union error indicates a bigger $l_q, l_r$ trade-off. $R_{\text{union}}$ is lower bounded by $\mathcal{R}_r$.

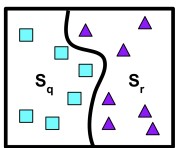

(a) $\mathcal{S}_q$ and $\mathcal{S}_r$ are disjoint and cover $X$

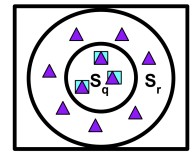

(b) $\mathcal{S}_r$ includes $\mathcal{S}_q$

Figure 2: $l_q - l_r$ trade-off visualization. Blue and purple points belong to $\mathcal{S}_q \subseteq X$ and $\mathcal{S}_r \subseteq X$.

Figure 3: Comparisons of union errors of two extreme cases. Note that $\mathcal{R}_r \leq \mathcal{R}_{\text{union}} \leq 1$. A larger union error has a more severe $l_q - l_r$ trade-off.

|  | $\mathcal{S}_q \cap \mathcal{S}_r = \emptyset \wedge \mathcal{S}_q \cup \mathcal{S}_r = X$ | $\mathcal{S}_q \subseteq \mathcal{S}_r$ |
|---|---|---|
| $\mathcal{R}_{\text{align}}$ | 1 - $\mathcal{R}_r$ | 0 |
| $\mathcal{R}_{\text{union}}$ | 1 | $\mathcal{R}_r$ (optimal) |

### 4.1 Certified training for multiple norms

Eq. 4 reveals that we need to minimize $\mathcal{R}_{\text{align}}$ by not only training with one kind of adversarial examples $x'_r$ since it will lead to a large $\mathcal{R}_{\text{align}}$ with more instances not robust against $l_q$ attack. To effectively combine the optimization of $l_q$ and $l_r$ ($q = 2, r = \infty$) certified training, inspired by the work (Tramer & Boneh, 2019; Madaan et al., 2021; Croce & Hein, 2022) on adversarial training for multiple norms, we propose the following methods:

1. **CURE-Joint**: optimizes $\mathcal{L}_{l_\infty}$ and $\mathcal{L}_{l_2}$ together: it takes the sum of two worst-case IBP losses with $l_\infty$ and $l_2$ examples using a convex combination of weights with hyperparameter $\alpha \in [0, 1]$.

$$\mathcal{L}_{Joint} = (1 - \alpha) \cdot \mathcal{L}_{l_\infty}(x, y, \epsilon_\infty, \tau_\infty) + \alpha \cdot \mathcal{L}_{l_2}(x, y, \epsilon_2, \tau_2)$$

2. **CURE-Max**: compares $\mathcal{L}_{l_2}$ and $\mathcal{L}_{l_\infty}$, selecting the higher IBP loss as the worse-case outcome. This approach acts as a *worst-case* defense, accounting for adversarial examples with the highest IBP loss across multiple perturbation types. The max loss $\mathcal{L}_{Max}$ is defined as:

$$\mathcal{L}_{Max} = \max_{p \in \{2, \infty\}} \max_{x' \in B_p(x, \epsilon_p - \tau_p)} \mathcal{L}_{\text{IBP}}(x, y, \epsilon_p, \tau_p)$$

3. **CURE-Random**: randomly partitions a batch of data $(\mathbf{x}, \mathbf{y}) \sim \mathcal{D}$ into equal sized blocks $(\mathbf{x}_1, \mathbf{y}_1)$ and $(\mathbf{x}_2, \mathbf{y}_2)$. For $(\mathbf{x}_1, \mathbf{y}_1)$, we calculate the $l_\infty$ worst-case IBP loss $\mathcal{L}_{l_\infty}$ with $l_\infty$ perturbations. For the other half $(\mathbf{x}_2, \mathbf{y}_2)$, similarly, we get the $l_2$ worst-case IBP loss by applying $l_2$ perturbations. After that, we optimize the **Joint** loss of these two with equal weights, as shown below. In this way, we reduce the time cost of propagating the bounds and generating the adversarial examples by $\frac{1}{2}$.

$$\mathcal{L}_{Random} = \mathcal{L}_{l_\infty}(\mathbf{x}_1, \mathbf{y}_2, \epsilon_\infty, \tau_\infty) + \mathcal{L}_{l_2}(\mathbf{x}_2, \mathbf{y}_2, \epsilon_2, \tau_2), \text{where } \mathbf{x} = \mathbf{x}_1 \cup \mathbf{x}_2, \mathbf{y} = \mathbf{y}_1 \cup \mathbf{y}_2$$

### 4.2 Unified and effective multi-norm certified training

The methods proposed above are still suboptimal as they fail to fully explore the relationship between worst-case IBP losses across different perturbations, certified training (CT), and natural training (NT). To address this, we introduce the following techniques to enhance the union robustness of **CURE**: (1) We derive an upper bound on the terms, which informs us to propose a bound alignment technique to mitigate the trade-off better, improving multi-norm robustness. (2) We analyze and connect certified and natural training to attain better union accuracy. (3) the first certified fine-tuning method to quickly improve union accuracy with pre-trained single-norm models (Table 1).

**Bound alignment (BA).** First, we aim to design *tight* upper bounds for different risk terms, leveraging the theory of classification-calibrated loss, which informs how to design methods to mitigate the $l_r - l_q$ tradeoff more efficiently. First, classification-calibrated surrogate loss is a surrogate loss $\mathcal{R}_\phi(f) := \mathbb{E}_{(x,y) \sim \mathcal{D}} \phi(f(x)y)$ designed to approximate the 0-1 loss, making it computationally efficient for optimization while maintaining a meaningful relationship with the true error (Zhang et al., 2019a). A loss is classification-calibrated if it ensures that any decision rule inconsistent with the Bayes optimal classifier has a strictly larger $\phi$-risk of the loss function $\phi$. This property is crucial for achieving optimal classification performance, and examples

include hinge loss, logistic loss, and exponential loss. Here, we show the binary IBP loss falls into this loss category.

**Lemma 4.1.** *Binary IBP loss is a logistic loss in the classification-calibrated surrogate loss family.*

*Proof.* We have binary $\mathcal{L}_{\mathrm{IBP}}(x, y, \epsilon_p) = \ln(1 + e^{\overline{o}_i - \underline{o}_y}), i \neq y$, which is a logistic loss.

**Upper bound.** Our following analysis provides a performance guarantee for minimizing the surrogate loss. We introduce a transformation $\psi$ of classification-calibrated losses. $\psi : [0, 1] \rightarrow [0, \infty)$ is defined as the convex conjugate of a function that lower bounds the gap between a modified entropy function (e.g., a surrogate loss like cross-entropy) and the standard Shannon entropy (Zhang et al., 2019a). This gap quantifies how well the surrogate loss approximates the true 0-1 classification error. The function $\psi$ is used to bound the difference between the union risk $\mathcal{R}_{\mathrm{union}}$ and the optimal risk under individual $\ell_r$ perturbations $\mathcal{R}_r^* := \min_f \mathcal{R}_r(f)$. It has desirable properties: $\psi$ is non-decreasing, convex, continuous on $[0, 1]$, and satisfies $\psi(0) = 0$. By Eq.4, we have $\mathcal{R}_{\mathrm{union}}(f) - \mathcal{R}_r^* = \mathcal{R}_r(f) - \mathcal{R}_r^* + \mathcal{R}_{\mathrm{align}}(f) \leq \psi^{-1}(\mathcal{R}_\phi(f) - \mathcal{R}_\phi^*) + \mathcal{R}_{\mathrm{align}}(f)$, where the inequality holds because $\phi$ is constructed from a classification-calibrated loss (Bartlett et al., 2006).

**Theorem 4.2.** *Let $\mathcal{R}_\phi(f) := \mathbb{E}\phi(f(x)y)$ and $\mathcal{R}_\phi^* := \min_f \mathcal{R}_\phi(f)$. Under Assumption 1 in Zhang et al. (2019a), with $\mathbb{E}$ taken over the data distribution, for any non-negative loss function $\phi$ such that $\phi(0) \geq 1$, any measurable $f : \mathcal{X} \rightarrow \mathbb{R}$, any probability distribution on $\mathcal{X} \times \{\pm 1\}$, IBP output bound differences from $f$ as $d_r(x) = \overline{o}_i - \underline{o}_y (i \neq y)$ for $l_r$ perturbations, and any $\lambda > 0$, we have*

$$\mathcal{R}_{union}(f) - \mathcal{R}_r^* \leq \psi^{-1}(\mathcal{R}_\phi(f) - \mathcal{R}_\phi^*) + \mathbb{E}(\phi(d_r(x)d_p(x)/\lambda), \overline{o}_i \leq \underline{o}_y \text{ for } d_r(x)).$$

The proof is in Appendix A.1, which sheds light on how we can further improve union-certified robustness. Algorithmically, we can extend the framework to the case of multi-class classifications by replacing $\phi$ with a multi-class calibrated loss $L(\cdot, \cdot)$ (Zhang et al., 2019a), such as cross-entropy, which ensures that minimizers of the surrogate risk align with those of the 0-1 loss. $\phi(d_r(x)d_p(x)/\lambda)$ indicates that we need to align the distributions between output bound differences of two perturbations, so Theorem 4.2 has a tighter upper bound. $\forall i \neq y, \overline{o}_i \leq \underline{o}_y$ means we need to regularize those bounds only on the *correctly predicted $l_r$ subsets* (Definition 4.3), meaning the subset $\gamma$ for which the lower bound computed with IBP of the correct class is higher than the upper bounds of other classes. Our analysis (Theorem 4.2) shows that the union-certified objective admits an upper bound decomposable into per-norm margin terms, and that misalignment between these margin distributions exacerbates the $\ell_q$–$\ell_r$ trade-off. Bound Alignment explicitly minimizes this mismatch, thereby tightening the upper bound on union error and preserving robustness across norms. In other words, aligning bound distributions reduces the variance between per-norm certification margins, preventing fine-tuning under one norm from disproportionately degrading another.

**Definition 4.3** (Correctly Certified $l_r$ Subset). At epoch $e$, given the perturbation size $\epsilon_r \in \mathbb{R}$ and model $f$, for a batch of data $(\mathbf{x}, \mathbf{y}) \sim \mathcal{D}$ of size $n$, we have the output upper and lower bounds computed by IBP for $l_r$ perturbations. We define a function $h$ for this procedure as $h(\mathbf{x}) = \{\overline{\mathbf{o}}_j, \underline{\mathbf{o}}_j\}_{j=0}^{j<n}$, where $\mathbf{o} = \{o_i\}_{i=0}^{i<k}$ is a vector of bounds for all classes. Then, the correctly certified subset $\gamma$ at the current step is defined as:

$$\forall j \in \gamma \text{ with } (\mathbf{x}_j, \mathbf{y}_j) \text{ and bounds } \{\overline{\mathbf{o}}_j = \{\overline{o}_i\}_{i=0}^{i<k}, \underline{\mathbf{o}}_j = \{\underline{o}_i\}_{i=0}^{i<k}\}, \text{ we have } \forall i \neq y_j, \overline{o}_i \leq \underline{o}_{y_j}.$$

For certified training, Gowal et al. (2018); Müller et al. (2022) optimize the model using bound differences $\{\overline{o}_i - \underline{o}_y\}_{i=0}^{i<k}$ ($y$ is the correct class). Inspired by Theorem 4.2, we align the bound differences $\{\{\overline{o}_i - \underline{o}_y\}_{i=0}^{i<k}\}_n$ of $l_r$ and $l_q$ CT outputs with a batch of $n$ samples, specifically on the correctly certified $l_q$ subset $\gamma$. Specifically, for each batch of data $(\mathbf{x}, \mathbf{y}) \sim \mathcal{D}$, we denote the bounds differences after softmax normalization for two perturbations as $d_q$ and $d_r$. Then, we select indices $\gamma$, according to Definition 4.3. We denote the size of the indices as $n_c \leq n$. We compute a KL-divergence loss over this set of samples using $KL(d_q[\gamma] \| d_r[\gamma])$ (Eq. 5). Intuitively, we aim to encourage $d_r[\gamma]$ and $d_q[\gamma]$ distributions to become close to each other, such that we gain more union robustness.

$$\mathcal{L}_{KL} = \frac{1}{n_c} \cdot \sum_{i=1}^{n_c} \sum_{j=0}^{k} d_q[\gamma[i]][j] \cdot \log\left(\frac{d_q[\gamma[i]][j]}{d_r[\gamma[i]][j]}\right) \tag{5}$$

Apart from the KL loss, we add another loss term using a Max-style approach in Eq. 6, since Max performs relatively well, as shown in Table 1. We also consider combining with Random/Joint losses if they lead to a better performance. Our final loss $\mathcal{L}_{\text{Scratch}}$ combines $\mathcal{L}_{KL}$ and $\mathcal{L}_{Max}$, via a hyper-parameter $\eta$, as shown in Eq. 7.

$$\mathcal{L}_{Max} = \max_{p\in\{2,\infty\}} \max_{x'\in B_p(x,\epsilon_p-\tau_p)} \mathcal{L}_{\text{IBP}}(x,y,\epsilon_p,\tau_p) \qquad (6) \qquad \mathcal{L}_{\text{Scratch}} = \mathcal{L}_{Max} + \eta \cdot \mathcal{L}_{KL} \qquad (7)$$

**Integrate NT into CT.** In the context of adversarial robustness, Jiang & Singh (2024) shows that there exist a useful portion of model updates in natural training, which can be extracted and integrated into adversarial training to improve adversarial robustness. Based on this, we propose a technique to integrate NT into CT, to enhance union-certified robustness. Specifically, for model $p^{(r)}$ at any epoch $r$, we examine the model updates of NT and CT over all samples from $\mathcal{D}$. The models $p_n^{(r)}$ and $p_c^{(r)}$ represent the results after one epoch of NT and CT, from the same initial model $p^{(r)}$. Then we compare the updates of the two $g_n = p_n^{(r)} - p^{(r)}$ and $g_c = p_c^{(r)} - p^{(r)}$. For a specific layer $l$, by comparing $g_n^l$ and $g_c^l$, we retain a portion of $g_n^l$ according to their cosine similarity score (Eq.8). Negative scores indicate that $g_n^l$ does not contribute to certified robustness, so we discard components with similarity scores $\leq 0$. The **GP** (Gradient Projection) operation, defined in Eq.9, projects $g_c^l$ towards $g_n^l$.

$$\cos(g_n^l, g_c^l) = \frac{g_n^l \cdot g_c^l}{\|g_n^l\|\|g_c^l\|} \qquad (8) \qquad \mathbf{GP}(g_n^l, g_c^l) = \begin{cases} \cos(g_n^l, g_c^l) \cdot g_n^l, & \cos(g_n^l, g_c^l) > 0 \\ 0, & \cos(g_n^l, g_c^l) \leq 0 \end{cases} \qquad (9)$$

Therefore, the total projected (useful) model updates $g_p$ coming from $g_n$ could be computed as Eq. 10. We use $\mathcal{M}$ to represent all layers of the current model update. The expression $\bigcup_{l\in\mathcal{M}}$ concatenates the useful natural model update components from all layers. A hyper-parameter $\beta$ is introduced to balance the contributions of $g_{GP}$ and $g_c$, as outlined in Eq.11. It is important to note that this projection procedure is applied only after the eps-annealing phase of certified training. The theoretical analysis of why connecting NT with CT works is discussed in Appendix A.2.

$$g_p = \bigcup_{l\in\mathcal{M}} \mathbf{GP}(g_n^l, g_c^l) \qquad (10) \qquad p^{(r+1)} = p^{(r)} + \beta \cdot g_p + (1-\beta) \cdot g_c \qquad (11)$$

**Quick certified fine-tuning.** In adversarial robustness, Croce & Hein (2022) shows that public models can be made more robust with only the application of fine-tuning, which reduces the computational cost significantly compared with training from scratch. In this work, we propose the first fine-tuning certified multi-norm robustness scheme **CURE-Finetune**. Starting from a single norm pre-trained model, we perform the bound alignment technique by optimizing $\mathcal{L}_{\text{Scratch}}$ for a few epochs. Because of the $l_q - l_r$ tradeoff, certifiably finetuning a $l_r$ pre-trained model on $l_q$ perturbations reduces $l_r$ robustness. Thus, we want to preserve more $l_r$ robustness when doing certified fine-tuning, which makes bound alignment useful here. By regularizing on the correctly certified $l_r$ subset with $\mathcal{L}_{\text{Scratch}}$, we can prevent losing more $l_r$ robustness when boosting $l_q$ robustness, which leads to better union accuracy. We note that **CURE-Finetune** can be adapted to any single-norm certifiably pre-trained models. As shown in Table 1, we can obtain a superior multi-norm certified robustness by performing quick fine-tuning on pre-trained $l_\infty$ models.

## 5    Experiment

In this section, we present and discuss the results of union, geometric, and patch robustness, as well as ablation studies on hyper-parameters for MNIST, CIFAR-10, and TinyImagenet experiments. Other ablation studies, visualizations, and algorithms of **CURE** can be found in Appendix C and E.

**Experimental Setup.**   For datasets, we use MNIST (LeCun et al., 2010) and CIFAR-10 (Krizhevsky et al., 2009) which both include 60K images with 50K and 10K images for training and testing, as well as TinyImageNet (Le & Yang, 2015) which consists of 200 object classes with 500 training images, 50 validation images, and 50 test images per class. We compare the following methods: 1. $l_\infty$: $l_\infty$ certified defense SABR (Müller et al., 2022), 2. $l_2$: $l_2$ certified defense based on SABR, 3. **CURE-Joint**: take a weighted sum of $l_2, l_\infty$ IBP losses. 4. **CURE-Max**: take the worst of $l_2, l_\infty$ IBP losses. 5. **CURE-Random**: randomly partitions the samples into two blocks, then applies the Joint loss with equal weights. 6. **CURE-Scratch**:

training from scratch with bound alignment and gradient projection techniques. 7. **CURE-Finetune**: robust fine-tuning with the bound alignment technique using $l_\infty$ pre-trained models. We use a 7-layer convolutional architecture CNN7, a standard architecture (Müller et al., 2022) for certified training. In Table 14, we compare our proposed $l_2$ defense with Hu et al. (2023), where we show our method outperforms the SOTA $l_2$ deterministic certified defense on CIFAR-10. We choose similar hyperparameters and training setup as Müller et al. (2022) for $l_\infty$ certified training. We select $\alpha = 0.5$, $l_2$ subselection ratio $\lambda_2 = 1e^{-5}$, $\beta = 0.5$, and $\eta = 2.0$ according to our ablation study results in Section 5.2 and Appendix C. For certified fine-tuning, we finetune 20% of the epochs of CURE-Scratch and are only performed on $l_\infty$ models as they generally have higher robust errors. Full implementation details are in Appendix B.

## 5.1 Main Results

**Evaluation.** We choose the common $\epsilon_\infty, \epsilon_2, \epsilon_1$ values used in the literature (Müller et al., 2022; Hu et al., 2023) to construct multi-norm regions. These include $(\epsilon_1 = 1.0, \epsilon_2 = 0.5, \epsilon_\infty = 0.1), (\epsilon_1 = 2.0, \epsilon_2 = 1.0, \epsilon_\infty = 0.3)$ for MNIST, $(\epsilon_1 = 0.5, \epsilon_2 = 0.25, \epsilon_\infty = \frac{2}{255}), (\epsilon_1 = 1.0, \epsilon_2 = 0.5, \epsilon_\infty = \frac{8}{255})$ for CIFAR-10 and $(\epsilon_1 = \frac{72}{255}, \epsilon_2 = \frac{36}{255}, \epsilon_\infty = \frac{1}{255})$ for TinyImageNet. We make sure the adversarial regions with sizes $\epsilon_\infty, \epsilon_1$ and $\epsilon_2$ do not include each other. We report the clean accuracy, certified accuracy against $l_1, l_2, l_\infty$ perturbations, union accuracy, and individual/average certified robustness against geometric transformations as well as patch attacks. Further, we use alpha-beta crown (Zhang et al., 2018) for certification on $l_2, l_\infty$ perturbations, FGV (Yang et al., 2022) for efficient certification of geometric transformations, and Chiang et al. (2020) for $2 \times 2$ patch attacks. Additional experiment results on CIFAR-100, varying epsilons for $l_p$ norms where we show our methods generalize to a wide choice of epsilons and ablation studies can be found in Appendix C.

| Dataset | $(\epsilon_\infty, \epsilon_2, \epsilon_1)$ | Methods | Clean | $l_\infty$ | $l_2$ | $l_1$ | Union |
|---|---|---|---|---|---|---|---|
| MNIST | (0.3, 1.0, 2.0) | $l_\infty$ | 98.7 | 92.1 | 69.6 | 38.9 | 38.5 |
| | | $l_2$ | 99.4 | 0.0 | 94.5 | 94.7 | 0.0 |
| | | CURE-Joint | 98.7 | 90.5 | 76.3 | 50.8 | 50.3 |
| | | CURE-Max | 98.7 | 91.1 | 76.2 | 47.2 | 46.5 |
| | | CURE-Random | 98.7 | 90.5 | 76.3 | 50.8 | 50.3 |
| | | CURE-Finetune | 98.5 | 90.1 | 83.5 | 64.0 | 63.2 |
| | | CURE-Scratch | 98.0 | 89.4 | 85.9 | 71.5 | **70.5** |
| CIFAR-10 | $(\frac{8}{255}, 0.5, 1.0)$ | $l_\infty$ | 51.8 | 36.3 | 6.0 | 3.8 | 3.5 |
| | | $l_2$ | 78.6 | 0.0 | 56.5 | 75.8 | 0.0 |
| | | CURE-Joint | 51.3 | 23.9 | 34.0 | 38.6 | 21.4 |
| | | CURE-Max | 51.5 | 33.9 | 19.5 | 21.6 | 16.8 |
| | | CURE-Random | 53.0 | 28.9 | 28.0 | 34.6 | 24.0 |
| | | CURE-Finetune | 40.2 | 30.2 | 30.8 | 34.8 | **29.3** |
| | | CURE-Scratch | 49.5 | 34.2 | 28.1 | 32.0 | 26.3 |
| TinyImagnet | $(\frac{1}{255}, \frac{36}{255}, \frac{72}{255})$ | $l_\infty$ | 28.3 | 19.4 | 19.4 | 12.9 | 12.9 |
| | | $l_2$ | 36.2 | 2.9 | 30.6 | 23.5 | 2.9 |
| | | CURE-Joint | 30.2 | 20.0 | 25.9 | 18.8 | 18.8 |
| | | CURE-Max | 29.6 | 21.8 | 23.5 | 18.2 | 18.2 |
| | | CURE-Random | 30.5 | 25.9 | 28.2 | 23.5 | **23.5** |
| | | CURE-Fintune | 28.1 | 21.2 | 21.8 | 18.2 | 16.6 |
| | | CURE-Scratch | 29.7 | 23.5 | 26.5 | 22.4 | 22.4 |

Table 1: Comparison of the clean accuracy, individual, and union certified accuracy (%). **CURE** consistently improves union accuracy compared with single-norm training with significant margins on all datasets. **CURE-Scratch** and **CURE-Finetune** outperform other methods in most cases.

**Union accuracy on MNIST, CIFAR-10, and TinyImagenet with CURE framework.** In Table 1, we show the results of clean accuracy and certified robustness against single and multi-norm with **CURE** on MNIST, CIFAR-10, and TinyImagenet. CURE-Joint, CURE-Max, and CURE-Random usually yield better union robustness than $l_2$ and $l_\infty$ certified training. Further, **CURE-Scratch** and **CURE-Finetune** consistently improve the union accuracy compared with other multi-norm methods with significant margins in most cases (20% for MNIST and 8% for CIFAR-10 experiments), showing the effectiveness of bound alignment and gradient projection techniques. Also, for quick fine-tuning, we show it is possible to quickly fine-tune a $l_\infty$ robust model with good union robustness using bound alignment, achieving SOTA union

| Configs | R(10) | R(2),Sh(2) | Sc(1),R(1), C(1),B(0.001) | Avg |
|---|---|---|---|---|
| $l_\infty$ | 27.8 | 33.2 | 23.3 | 28.1 |
| $l_2$ | 36.6 | 0.0 | 0.0 | 12.2 |
| CURE-Joint | **35.0** | **41.4** | **28.2** | **34.9** |
| CURE-Max | 33.7 | 39.0 | 23.3 | 32.0 |
| CURE-Random | 35.1 | 40.9 | 26.2 | 34.1 |
| CURE-Scratch | 34.2 | 39.6 | 24.9 | 32.9 |

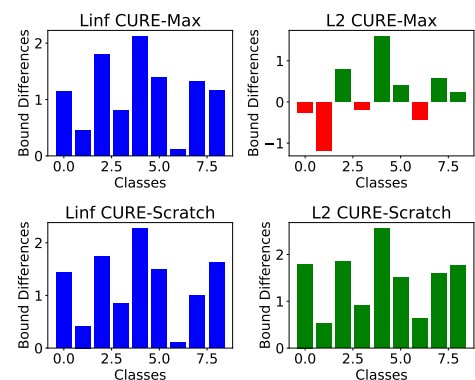

Figure 4: Comparison on **CURE** against geometric transformations for CIFAR-10 ($\epsilon_1 = 1.0, \epsilon_2 = 0.5, \epsilon_\infty = \frac{8}{255}$) experiment. We denote R($\varphi$) a rotation of $\pm\varphi$ degrees; $T_u(\Delta u)$ and $T_v(\Delta v)$ a translation of $\pm\Delta u$ pixels horizontally and $\pm\Delta v$ pixels vertically, respectively; Sc($\lambda$) a scaling of $\pm\lambda\%$; Sh($\gamma$) a shearing of $\pm\gamma\%$; C($\alpha$) a contrast change of $\pm\alpha\%$; and B($\beta$) a brightness change of $\pm\beta$. **CURE** improves the geometric certified robustness compared with single norm training. **CURE-Scratch** achieves the best average geometric transformation robustness.

Figure 5: CURE-Max and CURE-Scratch bound difference visualization. CURE-Scratch has a more similar bound distributions compared with CURE-Max.

accuracy on MNIST and CIFAR-10 experiments. More results on MNIST, CIFAR-10, and CIFAR-100 are available in Appendix C.

**Union accuracy on SST-2 dataset with BERT architectures.** Beyond vision tasks, we further apply CURE to a BERT-based model for text classification on the SST-2 dataset, operating in the latent space. As shown in Table 2, CURE-SCRATCH achieves the strongest robustness against both TextFooler (Jin et al., 2020) and PWWS (Ren et al., 2019) attacks. Since full certification for large-scale language models such as BERT is currently infeasible, we follow prior work and use strong empirical attacks as a proxy for union robustness evaluation. These results provide additional evidence that CURE improves generalized robustness beyond vision settings.

Table 2: Robust accuracy (%) on SST-2 under strong empirical attacks.

| Robust Acc | $\ell_\infty$-cert | $\ell_2$-cert | CURE-MAX | CURE-SCRATCH |
|---|---|---|---|---|
| SST-2 (PWWS (Ren et al., 2019)) | 16.8 | 15.2 | 24.0 | **28.4** |
| SST-2 (TextFooler (Jin et al., 2020)) | 9.4 | 10.0 | 15.6 | **17.6** |

**Robustness against unseen geometric and patch transformations.** Table 4 and Table 7 (in Appendix) compare **CURE** with single norm training against various geometric perturbations on MNIST and CIFAR-10 datasets. **CURE** outperforms single norm training on diverse geometric transformations (e.g., 6% for CIFAR-10 on average), leading to better *generalized certified robustness*. Also, **CURE-Scratch** has better geometric robustness than **CURE-Max** on both datasets, which reveals that bound alignment and gradient projection lead to better generalized certified robustness. In addition, in Table 3a, we display the certified robustness of **CURE** compared with single-norm baselines against patch $2 \times 2$ attacks. Our framework outperforms related baselines with 8.5% for MNIST and 16.0% for CIFAR-10, showing better *generalized certified robustness*. We hypothesize that many non-$l_p$ perturbations can be approximated or parameterized using $l_p$-bounded formulations, and improving $l_p$ robustness enhances robustness to such transformations - we find that CURE training achieves significantly higher bound overlap compared to single-norm models (Table 9). However, we also observe that a geometrically robust model lacks multi-norm robustness, as shown in Table 8 in Appendix.

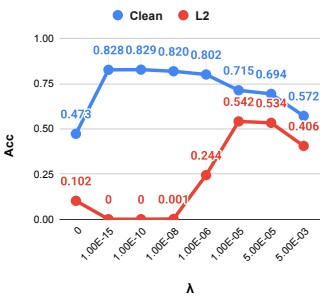

(a) $\lambda_2$: subselection ratio for $l_2$.

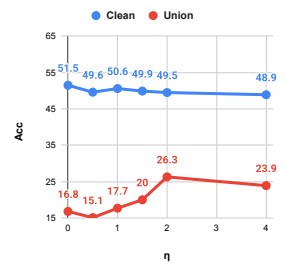

(b) $\eta$: weight for bound alignment.

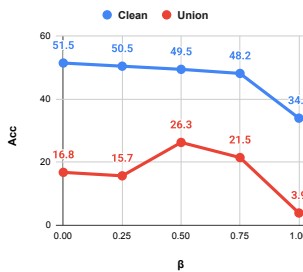

(c) $\beta$: hyper-parameter for GP.

Figure 6: Alabtion studies on $\lambda_2$, $\eta$ and $\beta$ hyper-parameters. According to the graphs, we choose $\lambda_2 = 1e^{-5}$, $\eta = 2.0$ and $\beta = 0.5$.

## 5.2 Ablation Study and Discussions

**Subselection ratio $\lambda$.** For $l_\infty$ certified training, we use the same $\lambda_\infty$ as in Müller et al. (2022). For $\lambda_2$, in Figure 6a, we show the $l_2$ certified robustness using varying $\lambda_2 \in [0, ..., 1e^{-2}]$ with $\epsilon_2 = 0.5$. According to Figure 6a, we choose $\lambda_2 = 1e^{-5}$.

**Bound alignment (BA) hyper-parameter $\eta$.** We perform CIFAR-10 ($\epsilon_\infty = \frac{8}{255}, \epsilon_2 = 0.5, \epsilon_1 = 1.0$) experiments with $\eta$ values in $[0.5, 1.0, 1.5, 2.0, 4.0]$. In Figure 6b, the clean accuracy generally drops as we have larger $\eta$ values, with union accuracy improving then dropping. We pick $\eta = 2.0$ with the best union accuracy for most experiments.

**Gradient projection (GP) hyper-parameter $\beta$.** Figure 6c displays the change of clean and union accuracy with choices of varying $\beta$ values on CIFAR-10 ($\epsilon_\infty = \frac{8}{255}, \epsilon_2 = 0.5, \epsilon_1 = 1.0$). CURE-Scratch is generally insensitive to $\beta$ values. Thus, we choose $\beta = 0.5$ for the experiments.

**Ablation study on BA and GP.** In Table 3b, we show the ablation study of BA and GP techniques on CIFAR-10 ($\epsilon_\infty = \frac{8}{255}, \epsilon_2 = 0.5, \epsilon_1 = 1.0$) experiment. BA and GP improve union accuracy by 6.8% and 2.7%, demonstrating the individual effectiveness of our proposed techniques.

| Methods | MNIST | CIFAR-10 |
|---|---|---|
| $l_\infty$ | 68.9 | 0.0 |
| $l_2$ | 0.0 | 0.0 |
| CURE-Joint | 68.5 | 0.2 |
| CURE-Max | 65.8 | 0.1 |
| CURE-Random | 72.8 | **16.0** |
| CURE-Scratch | **77.4** | 10.1 |

(a) Robust accuracy against $2 \times 2$ patch attacks on MNIST ($\epsilon_1 = 2.0, \epsilon_2 = 1.0, \epsilon_\infty = 0.3$) and CIFAR-10 ($\epsilon_1 = \frac{72}{255}, \epsilon_2 = \frac{36}{255}, \epsilon_\infty = \frac{1}{255}$) datasets. Results show CURE significantly outperforms single-norm training.

| | Clean | $l_\infty$ | $l_2$ | $l_1$ | Union |
|---|---|---|---|---|---|
| CURE-Max | 51.5 | 33.9 | 19.5 | 21.6 | 16.8 |
| +BA | 50.2 | 33.8 | 25.4 | 27.9 | 23.6 |
| +BA + GP | 49.5 | 34.2 | 28.1 | 32.0 | **26.3** |

(b) Ablations on BA and GP.

**Visualization of bound differences.** Figure 5 displays the bound differences $\{\underline{o}_y - \overline{o}_i\}_{i=0, i \neq y}^{i<k}$ of one example that is improved by **CURE-Scratch** (second row), compared with the **CURE-Max** (first row), from the CIFAR-10 ($\epsilon_\infty = \frac{8}{255}, \epsilon_2 = 0.5, \epsilon_1 = 1.0$) experiment. We use outputs from $\alpha, \beta$-CROWN. For $l_2$ perturbations (blue diagrams), **CURE-Scratch** consistently shows positive bound differences enabling robust union prediction, while **CURE-Max** has several negative ones (highlighted in red). The second-row distributions are more aligned than the first, showing that **CURE-Scratch** effectively aligns bound differences. This highlights the effectiveness of the bound alignment method. Additional visualizations are in Appendix C.

**Time cost of CURE.** The extra training costs of GP are small, taking $6, 24, 82$ seconds using a single NVIDIA A40 GPU on MNIST, CIFAR-10, and TinyImageNet datasets (Table 16), respectively. Compared with the total training cost of **CURE-Scratch**, it only accounts for $\sim 6\%$ of the total cost. For runtime

comparison of different methods with the same number of training epochs, we have a complete runtime analysis (Table 15) in Appendix D for the MNIST experiment. **CURE-Joint** has the largest cost among all methods. **CURE-Scratch** has a small extra time cost than **CURE-Max**, showing our proposed techniques have little additional cost.

**Limitations.** For $l_2$ certified training, we use a $l_\infty$ box instead of $l_2$ ball for bound propagation, which leads to more over-approximation and the potential loss of precision. Also, we notice drops in clean accuracy when training with **CURE** methods. BA and GP techniques lead to a slight decrease in clean accuracy in experiments. Further, our work does not claim to achieve universal certified robustness, but takes a step toward it by showing that multi-norm training offers broader certified robustness than single-norm or geometric-certified models (Table 8).

**Extension to other perturbation types.** Our framework is not inherently limited to the $\ell_2$–$\ell_\infty$ setting and can, in principle, be extended to $\ell_1$ or other $\ell_p$ norms. The key requirement is to identify the dominant tradeoff pairs between different $\ell_p$ threat models, which determines where robustness conflicts arise and where alignment is most beneficial. As shown in our prior work RAMP (Jiang & Singh, 2024) robustness tradeoffs are highly asymmetric across norms, and effective multi-norm training depends on explicitly targeting these critical tradeoff directions rather than treating all norms uniformly. That said, $\ell_1$ (and other $\ell_p$) certified training currently requires substantial additional engineering effort. In practice, $\ell_1$ certification involves looser bounds, higher computational cost, and less mature verification tooling compared to $\ell_2$ and $\ell_\infty$, which makes large-scale multi-norm certified training significantly more challenging. These limitations are largely orthogonal to our framework and stem from the current state of certified verification methods. As ceritification techniques general $\ell_p$ norms continue to improve, we expect our approach to extend naturally to these settings.

### 5.3 Discussion

We note that several components of our framework are inspired by prior work in adversarial (empirical) robustness (Tramer & Boneh, 2019; Zhang et al., 2019a; Madaan et al., 2021; Croce & Hein, 2022; Jiang & Singh, 2024). In particular, ideas such as joint training and gradient-based techniques are closely related to multi-objective optimization strategies commonly explored in adversarial training. These techniques are broadly applicable to robust learning, and our work builds upon these insights. At the same time, our framework is specifically developed in the context of certified robustness. In contrast to adversarial training methods that operate on logits or adversarial examples, our approach leverages certified bounds produced by the perturbations as the core optimization signal. As a result, our objective is to improve provable worst-case guarantees, rather than empirical robustness. Overall, while our method draws inspiration from prior robust training approaches, it adapts and extends these ideas to the certified setting by operating on certified bounds, leading to improvements in union-certified robustness.

## 6 Conclusion

We propose a framework **CURE** with multi-norm certified training methods for better union robustness. We establish a theoretical framework to analyze the tradeoff between perturbations, which inspires us to devise bound alignment, gradient projection, and robust certified fine-tuning techniques to enhance and facilitate the union-certified robustness. Extensive experiments on MNIST, CIFAR-10, and TinyImagenet show that **CURE** significantly improves union accuracy and robustness against geometric and patch transformations, paving the path to generalized certified robustness.

## 7 Acknowledgement

We sincerely thank the anonymous reviewers and the action editor for their insightful comments and constructive suggestions. This work was supported by NSF Grants No. CCF-2238079, CCF-2316233, CNS-2148583, Google Research Scholar Award, and an Open Philanthropy research grant.

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

# A  Proofs of the Theorems

In this section, we provide the proofs of the Theorems.

## A.1  Proof of Theorem 4.2

**Theorem 4.2 (restated).**  *Let $\mathcal{R}_\phi(f) := \mathbb{E}\phi(f(x)y)$ and $\mathcal{R}_\phi^* := \min_f \mathcal{R}_\phi(f)$. Under Assumption 1 in Zhang et al. (2019a), for any non-negative loss function $\phi$ such that $\phi(0) \geq 1$, any measurable $f : \mathcal{X} \to \mathbb{R}$, any probability distribution on $\mathcal{X} \times \{\pm 1\}$, IBP output bound differences from $f$ as $d_r(x) = \overline{o}_i - \underline{o}_y (i \neq y)$ for $l_r$ perturbations, and any $\lambda > 0$, we have*

$$\mathcal{R}_{union}(f) - \mathcal{R}_r^* \leq \psi^{-1}(\mathcal{R}_\phi(f) - \mathcal{R}_\phi^*) + \Pr[x_r' \in B_r(x, \epsilon_r), x_q' \in B_q(x, \epsilon_q), f(x_r')y > 0 \text{ and } f(x_q')y \leq 0]$$

$$\leq \psi^{-1}(\mathcal{R}_\phi(f) - \mathcal{R}_\phi^*) + \mathbb{E}\max_{\substack{x_r' \in B_r(x, \epsilon_r), \\ x_q' \in B_q(x, \epsilon_q)}} (\phi(f(x_r')f(x_p')/\lambda), f(x_r')y > 0)$$

$$\leq \psi^{-1}(\mathcal{R}_\phi(f) - \mathcal{R}_\phi^*) + \mathbb{E}(\phi(d_r(x)d_p(x)/\lambda), \overline{o}_i \leq \underline{o}_y \text{ for } d_r(x)).$$

*Proof.* By Eqn. equation 4, $\mathcal{R}_{\text{union}}(f) - \mathcal{R}_r^* = \mathcal{R}_r(f) - \mathcal{R}_r^* + \mathcal{R}_{\text{align}}(f) \leq \psi^{-1}(\mathcal{R}_\phi(f) - \mathcal{R}_\phi^*) + \mathcal{R}_{\text{align}}(f)$, where the last inequality holds because we choose $\phi$ as a classification-calibrated loss Bartlett et al. (2006). This leads to the first inequality.

Also, notice that

$$\Pr[x_r' \in B_r(x, \epsilon_r), x_q' \in B_q(x, \epsilon_q), f(x_r')y > 0 \text{ and } f(x_q')y \leq 0]$$

$$\leq \Pr[x_r' \in B_r(x, \epsilon_r), x_q' \in B_q(x, \epsilon_q), f(x_r')f(x_q') \leq 0, f(x_r')y > 0]$$

$$= \mathbb{E}\max_{x_r' \in B_r(x, \epsilon_r)} \max_{x_q' \in B_q(x, \epsilon_q)} (\mathbf{1}\{f(x_r') \neq f(x_q')\}, f(x_r')y > 0)$$

$$= \mathbb{E}\max_{x_r' \in B_r(x, \epsilon_r)} \max_{x_q' \in B_q(x, \epsilon_q)} (\mathbf{1}\{f(x_r')f(x_q')/\lambda < 0\}, f(x_r')y > 0)$$

$$\leq \mathbb{E}\max_{x_r' \in B_r(x, \epsilon_r)} \max_{x_q' \in B_q(x, \epsilon_q)} (\phi(f(x_r')f(x_q')/\lambda), f(x_r')y > 0)$$

$$\leq \mathbb{E}(\phi(d_r(x)d_p(x)/\lambda), \overline{o}_i \leq \underline{o}_y \text{ for } d_r(x)).$$

The last inequality holds because the adversarial loss is always upper-bounded by the IBP loss. Therefore, we get the second and third inequality in Theorem A.1. $\square$

### A.1.1  Multiclass extension

To characterize the robustness of a score function $f : \mathcal{X} \to \mathbb{R}^K$ for $K$-class classification, we define robust error under the threat model of $\ell_q$ perturbation:

$$\mathcal{R}_q(f) := \mathbb{E}_{(x,y) \sim \mathcal{D}}\left[\mathbf{1}\left\{\exists x_q' \in B_q(x, \epsilon_q) \text{ s.t. } f(x_q')_y \leq \max_{j \neq y} f(x_q')_j\right\}\right].$$

Here, $f(x)_y$ denotes the output logit for the true class $y$, and $\max_{j \neq y} f(x)_j$ is the maximum logit among all incorrect classes.

We define $\mathcal{R}_r(f)$ similarly to $\mathcal{R}_q(f)$, and without loss of generality, assume $\mathcal{R}_r(f) \geq \mathcal{R}_q(f)$. Then, we introduce the alignment error as the risk calculated on $x \in \mathcal{X}$ that are robust against $\ell_r$ attack but not robust against $\ell_q$ attack.

$$\mathcal{R}_{\text{align}}(f) := \mathbb{E}_{(x,y) \sim \mathcal{D}}\left[\mathbf{1}\left\{\begin{array}{l}\exists x_r' \in B_r(x, \epsilon_r), \ x_q' \in B_q(x, \epsilon_q), \text{ s.t.} \\ f(x_r')_y > \max_{j \neq y} f(x_r')_j \text{ and } f(x_q')_y \leq \max_{j \neq y} f(x_q')_j\end{array}\right\}\right].$$

We define union error as the risk calculated on $x \in \mathcal{X}$ that are either not robust against $\ell_q$ or $\ell_r$ attack. We have the following relationship:

$$\mathcal{R}_{\text{union}}(f) = \mathcal{R}_r(f) + \mathcal{R}_{\text{align}}(f).$$

**Theorem 4.2 (multiclass extension)** Let $\mathcal{R}_\phi(f) := \mathbb{E}\phi(f(x)_y - \max_{j \neq y} f(x)_j)$ and $\mathcal{R}_\phi^* := \min_f \mathcal{R}_\phi(f)$. Under the same assumption as in binary classification, for any non-negative loss function $\phi$ such that $\phi(0) \geq 1$, any measurable $f : \mathcal{X} \to \mathbb{R}^K$, any probability distribution on $\mathcal{X} \times \{1, \ldots, K\}$, and any $\lambda > 0$, we have

$$\mathcal{R}_{\text{union}}(f) - \mathcal{R}_r^* \leq \psi^{-1}(\mathcal{R}_\phi(f) - \mathcal{R}_\phi^*) + \mathcal{R}_{\text{align}}(f)$$
$$\leq \psi^{-1}(\mathcal{R}_\phi(f) - \mathcal{R}_\phi^*) + \mathbb{E}\Big[\phi\Big(\frac{d_r(x) \cdot d_q(x)}{\lambda}\Big) \cdot \mathbf{1}\{d_r(x) \leq 0\}\Big].$$

where we define the certified margin gaps (following binary case) as $d_r(x) = \max_{j \neq y} \overline{o}_j^{(r)} - \underline{o}_y^{(r)}, d_q(x) = \max_{j \neq y} \overline{o}_j^{(q)} - \underline{o}_y^{(q)}$ where $\underline{o}_y^{(r)}, \overline{o}_j^{(r)}$ are the IBP lower and upper bounds over $B_r(x, \epsilon_r)$, and similarly for $\ell_q$. Note: $d(x) \leq 0$ indicates certified robustness while $d(x) > 0$ indicates not certified robust.

*Proof.* $\mathcal{R}_{\text{union}}(f) - \mathcal{R}_r^* = \mathcal{R}_r(f) - \mathcal{R}_r^* + \mathcal{R}_{\text{align}}(f) \leq \psi^{-1}(\mathcal{R}_\phi(f) - \mathcal{R}_\phi^*) + \mathcal{R}_{\text{align}}(f)$, where the last inequality holds because we choose $\phi$ as a classification-calibrated loss for multiclass classification. This gives us the first inequality.

Now we bound $\mathcal{R}_{\text{align}}(f)$. Define the margin for any point as $m(x') = f(x')_y - \max_{j \neq y} f(x')_j$. The alignment condition becomes $\exists x'_r, x'_q$ such that $m(x'_r) > 0$ and $m(x'_q) \leq 0$.

Similar to binary classification,

$$\Pr[\exists x'_r \in B_r, x'_q \in B_q : m(x'_r) > 0 \text{ and } m(x'_q) \leq 0]$$
$$\leq \Pr[\exists x'_r \in B_r, x'_q \in B_q : m(x'_r) \cdot m(x'_q) \leq 0 \text{ and } m(x'_r) > 0]$$
$$= \mathbb{E} \max_{\substack{x'_r \in B_r(x, \epsilon_r), \\ x'_q \in B_q(x, \epsilon_q)}} \Big[\mathbf{1}\{m(x'_r) \neq m(x'_q)\} \cdot \mathbf{1}\{m(x'_r) > 0\}\Big]$$
$$= \mathbb{E} \max_{\substack{x'_r \in B_r(x, \epsilon_r), \\ x'_q \in B_q(x, \epsilon_q)}} \Big[\mathbf{1}\Big\{\frac{m(x'_r) \cdot m(x'_q)}{\lambda} < 0\Big\} \cdot \mathbf{1}\{m(x'_r) > 0\}\Big]$$
$$\leq \mathbb{E} \max_{\substack{x'_r \in B_r(x, \epsilon_r), \\ x'_q \in B_q(x, \epsilon_q)}} \Big[\phi\Big(\frac{m(x'_r) \cdot m(x'_q)}{\lambda}\Big) \cdot \mathbf{1}\{m(x'_r) > 0\}\Big]$$
$$\leq \mathbb{E}\Big[\phi\Big(\frac{d_r(x) \cdot d_q(x)}{\lambda}\Big) \cdot \mathbf{1}\{d_r(x) \leq 0\}\Big].$$

## A.2 Theory of connecting NT with CT

The proof for connecting NT with CT via gradient projection (GP) is very similar to what has been done in Jiang & Singh (2024), where authors analyze and compare the delta errors of two aggregation rules (standard training and training with GP). Delta errors are the indicators of convergences of different aggregation rules based on a mild assumption on the Lipschitz continuity of loss function gradients. GP leads to a smaller Delta error, which means GP results in a better convergence. The only difference in connecting NT with CT is that we use a different loss function compared with adversarial training, which makes the proof almost the same. One can refer to Jiang & Singh (2024) for the more detailed proof of GP.

## B  More training details

**Certified training for $l_2$ robustness.** We propose a new $l_2$ deterministic certified training method, inspired by SABR Müller et al. (2022). For the specified $\epsilon_2$ and $\tau_2$ values, we first generate adversarial examples by computing the gradient in the $l_2$ direction (Kim, 2020), then truncating the perturbation to lie within a slightly reduced $l_\infty$ ball $B_\infty(x, \epsilon_2 - \tau_2)$. After that, we propagate a smaller box region $B_\infty(x', \tau_2)$ using the IBP loss. The loss we optimize can be formulated as follows:

$$\mathcal{L}_{l_2}(x, y, \epsilon_2, \tau_2) = \max_{x' \in B_\infty(x, \epsilon_2 - \tau_2)} \mathcal{L}_{\text{IBP}}(x', y, \tau_2)$$

**Training details.** We mostly follow the hyper-parameter choices from Müller et al. (2022) for **CURE**. We include weight initialization and warm-up regularization from Shi et al. (2021). Further, we use ADAM (Kingma, 2014) with an initial learning rate of $1e^{-4}$, decayed twice with a factor of 0.2. For CIFAR-10, we train 160 and 180 epochs for $(\epsilon_\infty = \frac{2}{255}, \epsilon_2 = 0.25)$ and $(\epsilon_\infty = \frac{8}{255}, \epsilon_2 = 0.5)$, respectively. We decay the learning rate after 120 and 140, 140 and 160 epochs, respectively. For the TinyImagenet experiment, we use the same setting as the CIFAR-10 $(\epsilon_\infty = \frac{8}{255}, \epsilon_2 = 0.5)$ experiment. For the MNIST dataset, we train 70 epochs, decaying the learning rate after 50 and 60 epochs. For batch size, we set 128 for CIFAR-10 and TinyImagenet and 256 for MNIST. For all experiments, we first perform one epoch of standard training. Also, we anneal $\epsilon_\infty, \epsilon_2$ from 0 to their final values with 80 epochs for CIFAR-10 and TinyImagenet and 20 epochs for MNIST. We only apply GP after training with the final epsilon values. For certification, we verify 1000 examples on MNIST and CIFAR-10, as well as 170 examples on TinyImagenet. The values of all hyperparameters can be found in Table 4.

| | MNIST-small | MNIST-large | CIFAR-small | CIFAR-large | TinyImagenet |
|---|---|---|---|---|---|
| $\lambda_\infty$ | 0.4 | 0.6 | 0.1 | 0.7 | 0.4 |
| $\lambda_2$ | 1.00E-05 | 1.00E-05 | 1.00E-05 | 1.00E-05 | 1.00E-05 |
| Learning rate | 1.00E-04 | 1.00E-04 | 1.00E-04 | 1.00E-04 | 1.00E-04 |
| LR decay ratio | 0.2 | 0.2 | 0.2 | 0.2 | 0.2 |
| Training epochs | 70 | 70 | 160 | 180 | 180 |
| Decay epochs | 50, 60 | 50, 60 | 120, 140 | 140, 160 | 140, 160 |
| Batch size | 256 | 256 | 128 | 128 | 128 |
| $\alpha$ (CURE) | 0.5 | 0.5 | 0.5 | 0.5 | 0.5 |
| $\eta$ (CURE) | 2.0 | 0.5 | 2.0 | 2.0 | 2.0 |
| $\beta$ (CURE) | 0.5 | 0.5 | 0.5 | 0.5 | 0.5 |

Table 4: Training specifications of our main experiments on MNIST, CIFAR-10, and TinyImagenet.

**Certifications for evaluations on $l_1, l_2, l_\infty$ norms.** We evaluated our trained models using $\alpha, \beta$-CROWN (Zhang et al., 2018). Specifically, $\alpha, \beta$-CROWN employs an efficient linear bound propagation framework coupled with a branch-and-bound algorithm to certify the robustness of neural networks against adversarial attacks. It propagates bounds on network outputs layer-by-layer. These bounds are linear functions representing the range of potential values the network's output can take under a given set of input constraints. In addition, the branch-and-bound algorithm systematically divides the input space into smaller regions (branching) and computes tighter bounds on each subregion. $\alpha, \beta$-CROWN is versatile and supports various activation functions, including ReLU, sigmoid, and tanh, making it applicable to a wide range of neural network architectures. Also, it supports the certification on different $l_p(p = 1, 2, \infty)$ norms, which fits the goal of our CURE framework for multi-norm certified robustness.

## C  Other experiment results and ablation studies

**Additional experiment on MNIST ($\epsilon_\infty = 0.1$, $\epsilon_2 = 0.5$, $\epsilon_1 = 1.0$) and CIFAR-10 ($\epsilon_\infty = \frac{2}{255}$, $\epsilon_2 = 0.25$, $\epsilon_1 = 0.5$).** As shown in Table 5, our CURE-Scratch method achieves higher union-certified accuracy on both MNIST and CIFAR-10 compared to all baseline methods. This demonstrates that training from scratch with our proposed multi-norm certified training framework not only consistently outperforms single-norm approaches.

**Additional experiment on CIFAR-100 ($\epsilon_\infty = 2/255$, $\epsilon_2 = 0.25$, $\epsilon_1 = 0.5$).** As shown in Table 6, our CURE-Scratch method significantly improves union-certified accuracy on the CIFAR-100 dataset compared to all baseline methods. Specifically, CURE-Scratch reaches a union accuracy of 30.4%, outperforming CURE-Joint, CURE-MAX, and CURE-Random by substantial margins.

| Dataset | $(\epsilon_\infty, \epsilon_2, \epsilon_1)$ | Methods | Clean | $l_\infty$ | $l_2$ | $l_1$ | Union |
|---|---|---|---|---|---|---|---|
| MNIST | (0.1, 0.5, 1.0) | $l_\infty$ | 99.2 | 97.0 | 96.5 | 95.0 | 94.9 |
| | | $l_2$ | 99.5 | 2.6 | 98.6 | 98.0 | 2.6 |
| | | CURE-Joint | 99.2 | 97.6 | 97.9 | 97.5 | 97.2 |
| | | CURE-Max | 99.3 | 97.6 | 97.3 | 96.8 | 96.8 |
| | | CURE-Random | 99.2 | 97.3 | 97.2 | 97.1 | 96.9 |
| | | CURE-Finetune | 99.1 | 97.0 | 97.3 | 96.8 | 96.5 |
| | | CURE-Scratch | 99.2 | 97.5 | 98.0 | 97.9 | **97.5** |
| CIFAR-10 | $(\frac{2}{255}, 0.25, 0.5)$ | $l_\infty$ | 79.2 | 60.3 | 67.3 | 75.9 | 60.3 |
| | | $l_2$ | 82.1 | 5.8 | 71.3 | 81.7 | 5.8 |
| | | CURE-Joint | 79.4 | 56.2 | 68.1 | 77.1 | 56.2 |
| | | CURE-Max | 77.6 | 60.0 | 69.3 | 75.2 | 60.0 |
| | | CURE-Random | 78.4 | 59.0 | 68.5 | 76.9 | 58.9 |
| | | CURE-Finetune | 78.0 | 59.7 | 68.2 | 75.9 | 59.7 |
| | | CURE-Scratch | 76.0 | 61.2 | 67.7 | 74.6 | **61.2** |

Table 5: Comparison of the clean accuracy, individual, and union certified accuracy (%). **CURE** consistently improves union accuracy compared with single-norm training with significant margins on all datasets. **CURE-Scratch** and **CURE-Finetune** outperform other methods in most cases.

| | Clean | Linf (2/255) | L2 (0.25) | L1 (0.5) | Union |
|---|---|---|---|---|---|
| Linf (2/255) | 39.7 | 26.4 | 16.0 | 18.6 | 14.8 |
| L2 (0.25) | 54.3 | 2.4 | 37.4 | 47.4 | 2.4 |
| CURE-Joint | 42.5 | 28.0 | 26.8 | 32.4 | 26.0 |
| CURE-MAX | 40.7 | 26.8 | 22.8 | 29.4 | 22.6 |
| CURE-Random | 41.3 | 28.4 | 27.2 | 34.0 | 27.0 |
| CURE-Scratch | 40.4 | 30.6 | 31.4 | 36.2 | **30.4** |

Table 6: Multi-norm certified accuracy (%) on CIFAR100 dataset.

**Robustness against geometric transformations on CIFAR-10.** Table 7 displays the certified robustness against geometric transformations on CIFAR-10. **CURE** outperforms the single-norm baselines with significant margins. Also, we notice that CURE-Scratch improves CURE-Max, which indicates the effectiveness of bound alignment and gradient projection.

**CURE compares to models trained to be robust against geometric perturbations.** To evaluate whether geometric robustness generalizes to multi-norm robustness, we conducted additional experiments on CIFAR-10 using $(\epsilon_\infty = 2/255, \epsilon_2 = 0.25, \epsilon_1 = 0.5)$. We tested models trained with various geometric

| Configs | R(30) | $T_u(2),T_v(2)$ | Sc(5),R(5), C(5),B(0.01) | Sh(2),R(2),Sc(2), C(2),B(0.001) | Avg |
|---|---|---|---|---|---|
| $l_\infty$ | 54.6 | 20.9 | 82.5 | 95.6 | 63.4 |
| $l_2$ | 0.0 | 0.0 | 0.0 | 0.0 | 0.0 |
| CURE-Joint | **55.9** | 21.3 | 82.3 | **95.7** | 63.8 |
| CURE-Max | 50.1 | 20.7 | 80.2 | 94.8 | 61.5 |
| CURE-Random | 54.8 | 18.8 | 83.5 | 95.6 | 63.2 |
| CURE-Scratch | 51.0 | **24.3** | **85.5** | 95.1 | **64.0** |

Table 7: Comparison on **CURE** against geometric transformations for MNIST $(\epsilon_1 = 2.0, \epsilon_2 = 1.0, \epsilon_\infty = 0.3)$ experiment. **CURE** improves the generalized certified robustness significantly compared with single norm training.

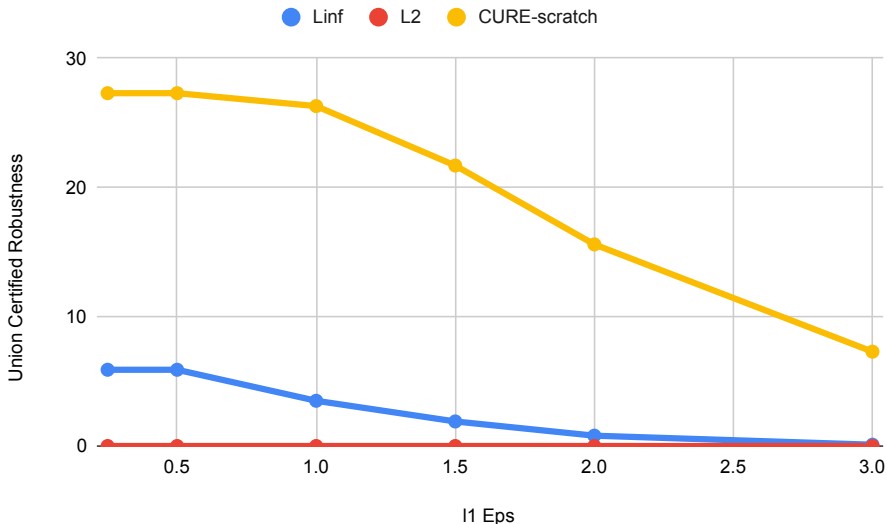

Figure 7: $l_\infty$, $l_2$ and CURE-scratch trained on CIFAR-10 union certified robustness analysis with varying $l_1$ epsilons.

transformations, including rotation $R(\varphi)$, translation $T_u(\Delta u), T_v(\Delta v)$, scaling $Sc(\lambda)$, shearing $Sh(\gamma)$, contrast $C(\alpha)$, and brightness $B(\beta)$, where the values denote the perturbation magnitudes (e.g., $R(10)$ applies up to $\pm 10°$ rotation). As shown in the table below, models trained with geometric perturbations Yang et al. (2022) achieve substantially lower union certified accuracy (e.g., 21.5%) compared to our CURE model (61.2%). This indicates that geometric robustness alone does not transfer well to multi-norm robustness, while our approach offers strong generalization across diverse norm-bounded threats.

| Method | $\ell_\infty$ | $\ell_2$ | $\ell_1$ | Union |
|---|---|---|---|---|
| CGT: R(10) | 1.6 | 22.8 | 38.0 | 1.6 |
| CGT: R(2), Sh(2) | 32.2 | 22.8 | 38.0 | 18.1 |
| CGT: Sc(1), R(1), C(1), B(0.001) | 33.2 | 22.8 | 38.0 | 21.5 |
| Ours | 61.2 | 67.7 | 74.6 | **61.2** |

Table 8: Comparison of CGT (Yang et al., 2022) versus our CURE model on CIFAR-10.

**Comparing bound overlap across models.** In Table 9, we compare single-norm and multi-norm trained models in terms of their bound overlap with CGT models. For fairness, we compute the maximum overlap across each batch and normalize the bound outputs. The results show that CURE-Scratch achieves substantially higher overlap than the $\ell_\infty$ certified baseline, highlighting its stronger alignment and generalization across perturbation types.

**$l_\infty$, $l_2$ and CURE-scratch trained on CIFAR-10 ($\epsilon_\infty = 8/255$, $\epsilon_2 = 0.5$, $\epsilon_1 = 1.0$) union certified robustness analysis with varying $l_\infty$, $l_2$, and $l_1$ epsilons.** We evaluate the certified robustness of our trained $l_\infty$, $l_2$, and CURE-Scratch models across a range of perturbation sizes under $l_1$, $l_2$, and $l_\infty$ norms. This comprehensive evaluation reveals that CURE-Scratch consistently outperforms the single-norm trained models across all tested settings. The results highlight the effectiveness of our approach in achieving strong multi-norm certified robustness, demonstrating that CURE-Scratch not only generalizes better across different norms but also maintains superior certification performance under varying attack strengths.

**The overlapping of $l_\infty$ and $l_2$ balls.** A reasonable misconception is that because $l_\infty$ and $l_2$ balls contain some overlap, training for robustness in one norm will sufficiently account for the weakness in the other.

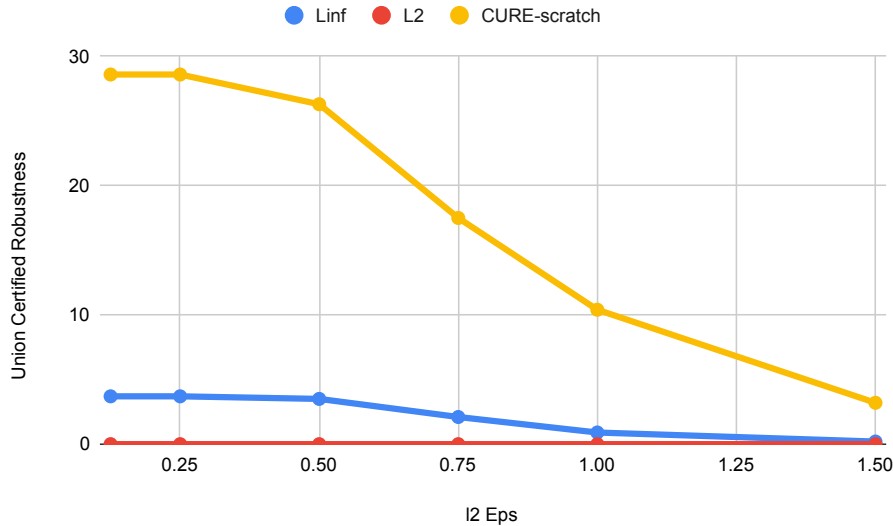

Figure 8: $l_\infty$, $l_2$ and CURE-scratch trained on CIFAR-10 union certified robustness analysis with varying $l_2$ epsilons.

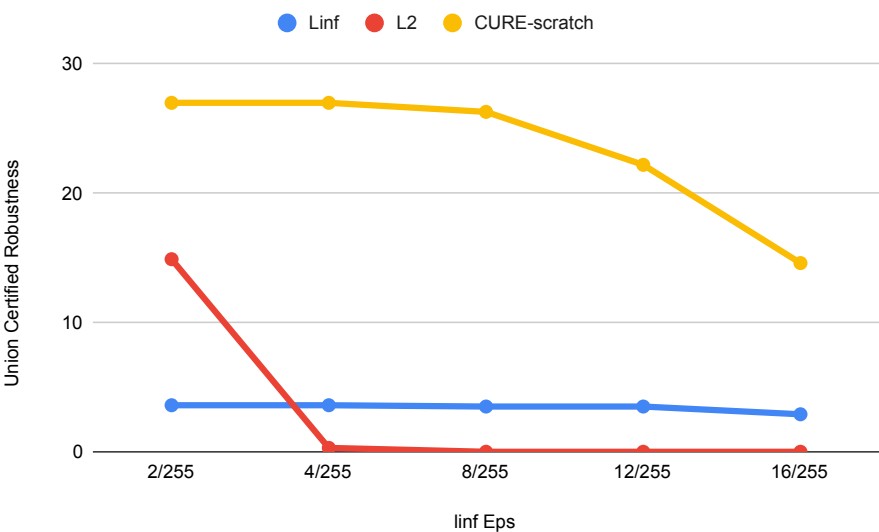

Figure 9: $l_\infty$, $l_2$ and CURE-scratch trained on CIFAR-10 union certified robustness analysis with varying $l_\infty$ epsilons.

|  | R(10) R(2), Sh(2) | Sc(1), R(1), C(1), B(0.001) |
|---|---|---|
| Linf | 0.141    0.723 | 0.791 |
| CURE-Scratch | **0.277**    **0.789** | **0.892** |

Table 9: Comparison of multi-norm (CURE-Scratch) versus single-norm certified trained models on the bound overlap with the CGT model.

Besides choices of $l_\infty$ and $l_2$ that completely overlap each other, the true regions of successful attacks have a significant mismatch across different norms.

To illustrate the mismatch between $l_\infty$ and $l_2$ regions, it suffices to show the existence of successful attacks that lie further enough from the original data input such that they are not covered by the other norm ball. We ran PGD 19968 times through a full testing run of a naturally trained model in the MNIST setting ($\epsilon_\infty = 0.1, \epsilon_2 = 0.5$) with the following results:

| % of $\ell_\infty$ attacks not in $\ell_2$ ball | % of $\ell_2$ attacks not in $\ell_\infty$ ball |
|---|---|
| 100.00% (9984/9984) | 98.95% (9879/9984) |

Table 10: Comparison of $\ell_\infty$ and $\ell_2$ attack coverage.

Of course, PGD may not find the absolute strongest adversarial examples in each ball. That only makes the mismatch claim stronger, because if PGD can find adversarial examples outside the other norm's ball, more attacks almost certainly exist in those regions as well.

**Comparison of $l_2$ certified training and PGD training.** Table 11 shows the $l_2$ certified robustness comparison between certified training and PGD training. The results demonstrate that determinist-certified training greatly improves the certified robustness.

| $l_2$ certified robustness | MNIST-large | CIFAR-small | CIFAR-large |
|---|---|---|---|
| Certified training | **94.5** | **71.2** | **56.6** |
| PGD training | 74.3 | 23.3 | 10.2 |

Table 11: Comparison on $l_2$ certified robustness between certified and PGD training.

**Comparison on empirical robustness on SST-2 using BERT model.** Moreover, our advances in certified robustness also translate into strong empirical robustness on transformer-based language models. As shown in Table 12, when applied to a BERT model on the SST-2 dataset, CURE-Scratch consistently outperforms FLAT (Chen & Ji, 2022), a state-of-the-art empirical adversarial training method, under both PWWS and TextFooler attacks. This result is particularly notable because CURE is designed around certified objectives rather than direct optimization against empirical attacks, yet it achieves substantially higher empirical robustness. These findings further indicate that our approach is not a straightforward or trivial adaptation of adversarial training techniques, but instead introduces optimization principles that generalize beyond the certified setting.

**Hyper-parameter $\alpha$ for Joint certified training.** As shown in Table 13, we test the changing of $l_\infty$, $l_2$, and union accuracy with different $\alpha$ values in $[0, 0.25, 0.5, 0.75, 1.0]$ on MNIST ($\epsilon_\infty = 0.1, \epsilon_2 = 0.5$) experiments. We observe that $\alpha = 0.5$ has the best union accuracy and is generally a good choice for our experiments by balancing the two losses.

**Comparison of $l_2$ certified robustness on $l_2$ deterministic certified training methods.** In Table 14, we compare our proposed $l_2$ certified defense with SOTA $l_2$ certified defense Hu et al. (2023) on CIFAR-10 with $\epsilon_2 = 0.25$ and $0.5$. Hu et al. (2023)'s model is a Lipchitz model and thus can report the certified robust accuracy directly after forward pass, whereas CURE requires an external certification procedure. The results

Table 12: Empirical robustness (%) on SST-2 for a BERT model under text attacks.

| Robust Accuracy | FLAT (Chen & Ji, 2022) | CURE-Scratch |
|---|---|---|
| SST-2 (PWWS) | 14.6 | **28.4** |
| SST-2 (TextFooler) | 12.4 | **17.6** |

| $\alpha$ | 0.0 | 0.25 | 0.5 | 0.75 | 1.0 |
|---|---|---|---|---|---|
| Clean | 99.2 | 99.2 | 99.3 | 99.2 | 99.5 |
| $l_\infty$ | 97.7 | 97.7 | 97.5 | 97.2 | 2.0 |
| $l_2$ | 96.9 | 95.6 | 97.4 | 95.9 | 98.7 |
| Union | 96.9 | 95.6 | **97.1** | 95.8 | 2.0 |

Table 13: Ablation study on Joint training hyper-parameter $\alpha$.

show that our proposed $l_2$ deterministic certified training method improves over $l_2$ robustness by $2 \sim 4\%$ compared with the SOTA method.

| $\epsilon_2$ | 0.25 | 0.5 |
|---|---|---|
| Hu et al. (2023) | 69.5 | 52.2 |
| Ours | **71.2** | **56.6** |

Table 14: Comparison of $l_2$ certified accuracy: our proposed $l_2$ certified training consistently outperforms Hu et al. (2023) by $2 \sim 4\%$.

**More visualizations on bound differences.** We plot the bound difference examples from alpha-beta-crown on MNIST, CIFAR-10, and TinyImagenet datasets, where the negative bound differences are colored in red. As shown in Figure 10, 11, 12, we compare CURE-Scratch (second row) with CURE-Max (first row), with bound differences against $l_\infty$ and $l_2$ perturbations colored in blue and green, respectively. CURE-Scratch produces all positive bound differences, leading to unionly robust predictions; CURE-Max is not unionly robust due to some negative bound differences. Also, we observe that CURE-Scratch successfully brings $l_q, l_r$ bound difference distributions close to each other compared with CURE-Max in many cases, which confirms the effectiveness of our bound alignment technique.

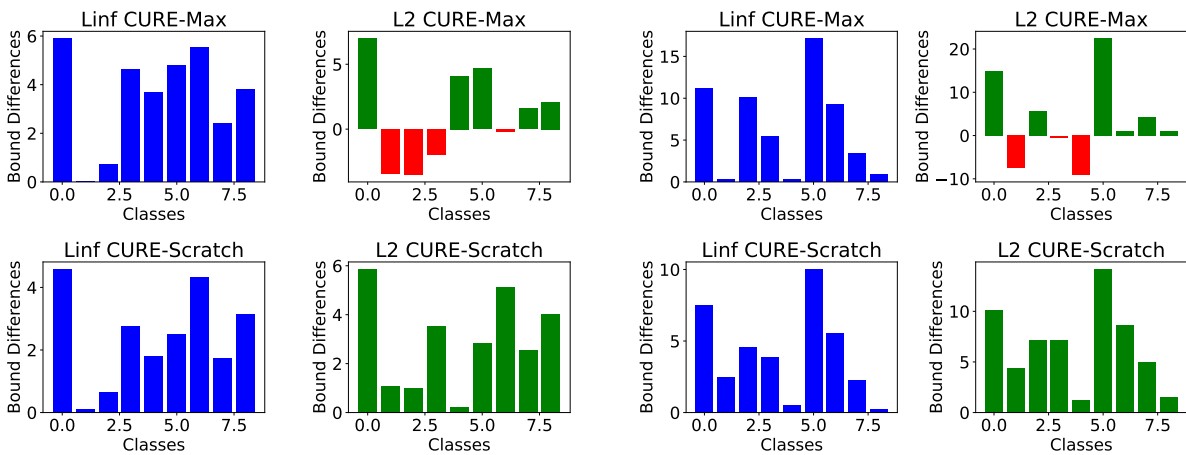

Figure 10: Bound difference visualizations on MNIST ($\epsilon_\infty = 0.3, \epsilon_2 = 1.0$) experiments.

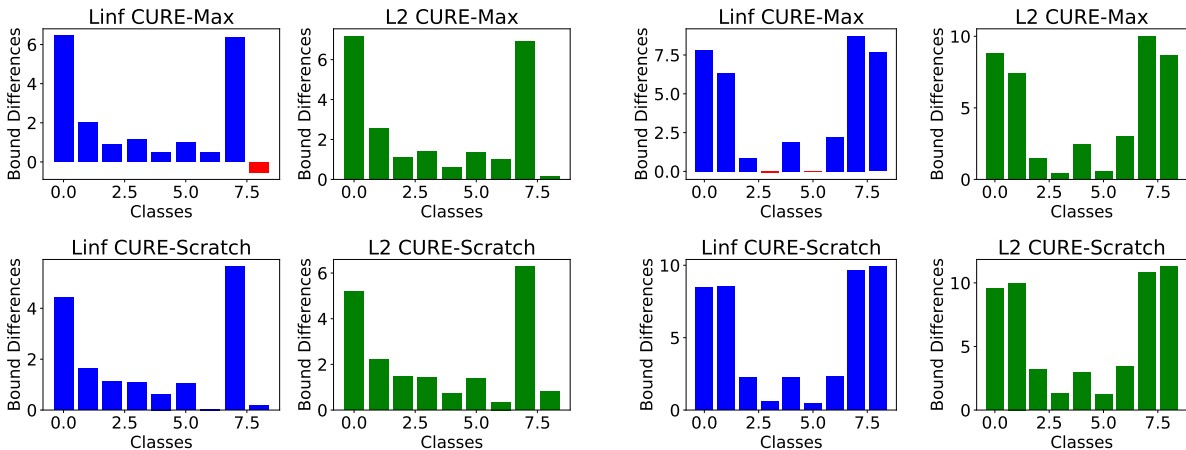

Figure 11: Bound difference visualizations on CIFAR-10 ($\epsilon_\infty = \frac{2}{255}, \epsilon_2 = 0.25$) experiments.

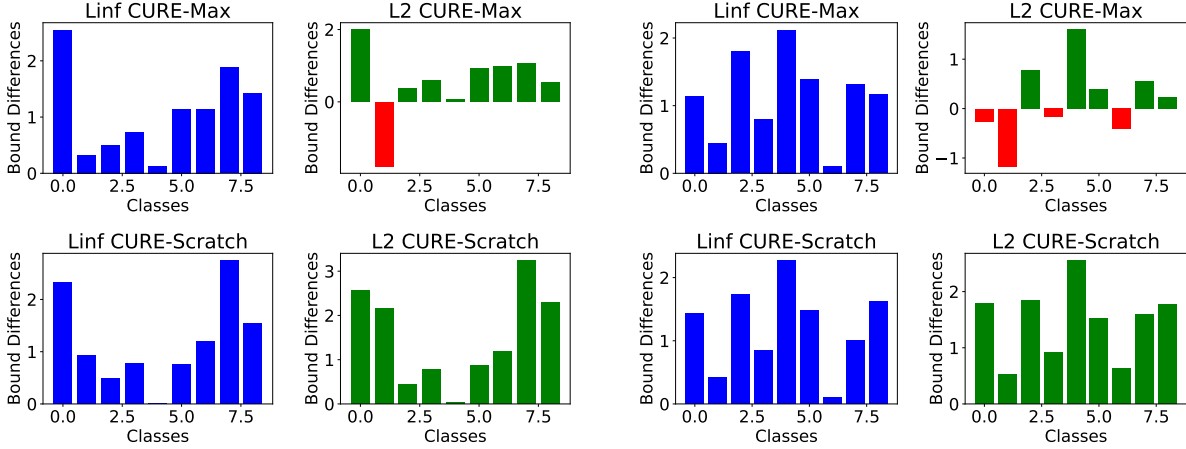

Figure 12: Bound difference visualizations on CIFAR-10 ($\epsilon_\infty = \frac{8}{255}, \epsilon_2 = 0.5$) experiments.

# D    Runtime Analysis

This section provides the runtime per training epoch for all methods on MNIST ($\epsilon_\infty = 0.1, \epsilon_2 = 0.75$) experiments and runtime per training epoch of CURE-Scratch with ablation studies on GP for MNIST, CIFAR10, and TinyImagenet experiments. We evaluate all the methods on a single A40 Nvidia GPU with 40GB memory and the runtime is reported in seconds (s).

**Runtime for different methods on MNIST experiments.** In Table 15, we show the time in seconds (s) per training epoch for single norm training ($l_\infty$ and $l_2$), CURE-Joint, CURE-Max, CURE-Random, CURE-Scratch, and CURE-Finetune methods. CURE-Finetune has a relatively small training cost compared with other methods and CURE-Joint has the highest time cost (around two times of other methods) per epoch. The results indicate the efficiency of training with CURE-Scratch/Finetune.

**Runtime for CURE-Scratch on MNIST, CIFAR10, and TinyImagenet datasets.** In Table 16, we show the runtime per training epoch using **CURE-Scratch** on MNIST, CIFAR10, and TinyImagenet datasets with and without GP operations. We see that the GP operation's cost is small compared with the whole training procedure, accounting for around 6% of the whole training time.

| Methods | Runtime (s) |
|---|---|
| $l_\infty$ | 89 |
| $l_2$ | 82 |
| CURE-Joint | 155 |
| CURE-Max | 147 |
| CURE-Random | 101 |
| CURE-Finetune | 148 |
| CURE-Scratch | 153 |

Table 15: Runtime for all methods on MNIST ($\epsilon_\infty = 0.1, \epsilon_2 = 0.5$) experiment per epoch in seconds.

| | MNIST | CIFAR-10 | TinyImagenet |
|---|---|---|---|
| w/o GP | 148 | 390 | 952 |
| with GP | 154 | 414 | 1036 |

Table 16: Runtime for CURE-Scratch on MNIST, CIFAR10, and TinyImagenet datasets.

## E  Algorithms

In this section, we present the algorithms of **CURE** framework. Algorithm 1 illustrates how to get propagation region for both $l_2$ and $l_\infty$ perturbations. Algorithm 2, 3, 4, 5 refer to algorithms of CURE-Joint, CURE-Max, CURE-Random, and CURE-Scratch/Finetune, respectively. Algorithm 6 is the procedure of performing GP after one epoch of natural and certified training (could be any of Algorithm 2, 3, 4, 5).

---

**Algorithm 1** get_propagation_region for $l_\infty$ and $l_2$ perturbations

---

**Require:** Neural network $f$, input $\boldsymbol{x}$, label $t$, perturbation radius $\epsilon$, subselection ratio $\lambda$, step size $\alpha$, step number $n$, attack types $\in \{l_\infty, l_2\}$

**Ensure:** Center $\boldsymbol{x}'$ and radius $\tau$ of propagation region $\mathcal{B}^\tau(\boldsymbol{x}')$

  $(\underline{\boldsymbol{x}}, \overline{\boldsymbol{x}}) \leftarrow \text{clamp}((\boldsymbol{x} - \epsilon, \boldsymbol{x} + \epsilon), 0, 1)$       // Get bounds of input region

  $\boldsymbol{\tau} \leftarrow \lambda/2 \cdot (\overline{\boldsymbol{x}} - \underline{\boldsymbol{x}})$       // Compute propagation region size $\tau$

  $\boldsymbol{x}_0^* \leftarrow \text{Uniform}(\underline{\boldsymbol{x}}, \overline{\boldsymbol{x}})$       // Sample PGD initialization

  **for** $i = 0 \ldots n - 1$ **do**       // Do $n$ PGD steps

    **if** attack $= l_\infty$ **then**       // Find examples with $l_\infty$ gradient direction

      $\boldsymbol{x}_{i+1}^* \leftarrow \boldsymbol{x}_i^* + \alpha \cdot \epsilon \cdot \text{sign}(\nabla_{\boldsymbol{x}_i^*} \mathcal{L}_{\text{CE}}(f(\boldsymbol{x}_i^*), t))$

      $\boldsymbol{x}_{i+1}^* \leftarrow \text{clamp}(\boldsymbol{x}_{i+1}^*, \underline{\boldsymbol{x}}, \overline{\boldsymbol{x}})$

    **end if**

    **if** attack $= l_2$ **then**       // Find examples with $l_2$ gradient direction

      $\boldsymbol{x}_{i+1}^* \leftarrow \boldsymbol{x}_i^* + \alpha \cdot \frac{\nabla_{\boldsymbol{x}_i^*} \mathcal{L}_{\text{CE}}(f(\boldsymbol{x}_i^*), \boldsymbol{y})}{\|\nabla_{\boldsymbol{x}_i^*} \mathcal{L}_{\text{CE}}(f(\boldsymbol{x}_i^*), \boldsymbol{y})\|_2}$

      $\delta \leftarrow \frac{\epsilon}{\|\boldsymbol{x}_{i+1}^* - \boldsymbol{x}\|_2} \cdot (\boldsymbol{x}_{i+1}^* - \boldsymbol{x})$

      $\boldsymbol{x}_{i+1}^* \leftarrow \text{clamp}(\boldsymbol{x} + \delta, \underline{\boldsymbol{x}}, \overline{\boldsymbol{x}})$

    **end if**

  **end for**

  $\boldsymbol{x}' \leftarrow \text{clamp}(\boldsymbol{x}_n^*, \underline{\boldsymbol{x}} + \tau, \overline{\boldsymbol{x}} - \tau)$       // Ensure that $\mathcal{B}^\tau(\boldsymbol{x}')$ will lie fully in $\mathcal{B}^\epsilon(\boldsymbol{x})$

  **return** $\boldsymbol{x}' t, \tau$

---

---

**Algorithm 2** CURE-Joint Training Epoch

---

**Require:** Neural network $f_\theta$, training set $(\boldsymbol{X}, \boldsymbol{T})$, perturbation radius $\epsilon_2$ and $\epsilon_\infty$, subselection ratio $\lambda_\infty$ and $\lambda_2$, learning rate $\eta$, $\ell_1$ regularization weight $\ell_1$, loss balance factor $\alpha$

    **for** $(\boldsymbol{x}, t) = (\boldsymbol{x}_0, t_0) \ldots (\boldsymbol{x}_b, t_b)$ **do**          // Sample batches $\sim (\boldsymbol{X}, \boldsymbol{T})$

        $(\boldsymbol{x}'_\infty, \tau_\infty) \leftarrow$ get_propagation_region (attack $= l_\infty$) // Refer to Algorithm 1

        $(\boldsymbol{x}'_2, \tau_2) \leftarrow$ get_propagation_region (attack $= l_2$)

        $\mathcal{B}^{\tau_\infty}(\boldsymbol{x}'_\infty) \leftarrow \text{Box}(\boldsymbol{x}'_\infty, \tau_\infty)$          // Get box with midpoint $\boldsymbol{x}'_\infty, \boldsymbol{x}'_2$ and radius $\tau_\infty, \tau_2$

        $\mathcal{B}^{\tau_2}(\boldsymbol{x}'_2) \leftarrow \text{Box}(\boldsymbol{x}'_2, \tau_2)$

        $\boldsymbol{u}_{y^\triangle_\infty} \leftarrow$ get_upper_bound$(f_\theta, \mathcal{B}^{\tau_\infty}(\boldsymbol{x}'_\infty))$          // Get upper bound $\boldsymbol{u}_{y^\triangle_\infty}, \boldsymbol{u}_{y^\triangle_2}$ on logit differences

        $\boldsymbol{u}_{y^\triangle_2} \leftarrow$ get_upper_bound$(f_\theta, \mathcal{B}^{\tau_2}(\boldsymbol{x}'_2))$          // based on IBP

        $\text{loss}_{l_\infty} \leftarrow \mathcal{L}_{\text{CE}}(\boldsymbol{u}_{y^\triangle_\infty}, t)$

        $\text{loss}_{l_2} \leftarrow \mathcal{L}_{\text{CE}}(\boldsymbol{u}_{y^\triangle_2}, t)$

        $\text{loss}_{\ell_1} \leftarrow \ell_1 \cdot$ get_$\ell_1$_norm$(f_\theta)$

        $\text{loss}_{tot} \leftarrow (1 - \alpha) \cdot \text{loss}_{l_\infty} + \alpha \cdot \text{loss}_{l_2} + \text{loss}_{\ell_1}$

        $\theta \leftarrow \theta - \eta \cdot \nabla_\theta \text{loss}_{tot}$          // Update model parameters $\theta$

    **end for**

---

**Algorithm 3** CURE-Max Training Epoch

---

**Require:** Neural network $f_\theta$, training set $(\boldsymbol{X}, \boldsymbol{T})$, perturbation radius $\epsilon_2$ and $\epsilon_\infty$, subselection ratio $\lambda_\infty$ and $\lambda_2$, learning rate $\eta$, $\ell_1$ regularization weight $\ell_1$

    **for** $(\boldsymbol{x}, t) = (\boldsymbol{x}_0, t_0) \ldots (\boldsymbol{x}_b, t_b)$ **do**          // Sample batches $\sim (\boldsymbol{X}, \boldsymbol{T})$

        $(\boldsymbol{x}'_\infty, \tau_\infty) \leftarrow$ get_propagation_region (attack $= l_\infty$) // Refer to Algorithm 1

        $(\boldsymbol{x}'_2, \tau_2) \leftarrow$ get_propagation_region (attack $= l_2$)

        $\mathcal{B}^{\tau_\infty}(\boldsymbol{x}'_\infty) \leftarrow \text{Box}(\boldsymbol{x}'_\infty, \tau_\infty)$          // Get box with midpoint $\boldsymbol{x}'_\infty, \boldsymbol{x}'_2$ and radius $\tau_\infty, \tau_2$

        $\mathcal{B}^{\tau_2}(\boldsymbol{x}'_2) \leftarrow \text{Box}(\boldsymbol{x}'_2, \tau_2)$

        $\boldsymbol{u}_{y^\triangle_\infty} \leftarrow$ get_upper_bound$(f_\theta, \mathcal{B}^{\tau_\infty}(\boldsymbol{x}'_\infty))$          // Get upper bound $\boldsymbol{u}_{y^\triangle_\infty}, \boldsymbol{u}_{y^\triangle_2}$ on logit differences

        $\boldsymbol{u}_{y^\triangle_2} \leftarrow$ get_upper_bound$(f_\theta, \mathcal{B}^{\tau_2}(\boldsymbol{x}'_2))$          // based on IBP

        $\text{loss}_{l_\infty} \leftarrow \mathcal{L}_{\text{CE}}(\boldsymbol{u}_{y^\triangle_\infty}, t)$

        $\text{loss}_{l_2} \leftarrow \mathcal{L}_{\text{CE}}(\boldsymbol{u}_{y^\triangle_2}, t)$

        $\text{loss}_{Max} \leftarrow max(\text{loss}_{l_\infty}, \text{loss}_{l_2})$          // We select the largest $l_{p \in [2, \infty]}$ loss for each sample

        $\text{loss}_{\ell_1} \leftarrow \ell_1 \cdot$ get_$\ell_1$_norm$(f_\theta)$

        $\text{loss}_{tot} \leftarrow \text{loss}_{Max} + \text{loss}_{\ell_1}$

        $\theta \leftarrow \theta - \eta \cdot \nabla_\theta \text{loss}_{tot}$          // Update model parameters $\theta$

    **end for**

---

---

**Algorithm 4** CURE-Random Training Epoch

---

**Require:** Neural network $f_\theta$, training set $(\boldsymbol{X}, \boldsymbol{T})$, perturbation radius $\epsilon_2$ and $\epsilon_\infty$, subselection ratio $\lambda_\infty$ and $\lambda_2$, learning rate $\eta$, $\ell_1$ regularization weight $\ell_1$

   **for** $(\boldsymbol{x}, t) = (\boldsymbol{x}_0, t_0) \ldots (\boldsymbol{x}_b, t_b)$ **do**        // Sample batches $\sim (\boldsymbol{X}, \boldsymbol{T})$

     $(\boldsymbol{x}_1, \boldsymbol{x}_2), (t_1, t_2) \leftarrow \text{partition}(\boldsymbol{x}, t)$       // Randomly partition inputs into two blocks

                                                      // Apply Algorithm 1

     $(\boldsymbol{x}'_\infty, \tau_\infty) \leftarrow \text{get\_propagation\_region}(\boldsymbol{x}_1, t_1, \text{attack} = l_\infty)$

     $(\boldsymbol{x}'_2, \tau_2) \leftarrow \text{get\_propagation\_region}(\boldsymbol{x}_2, t_2, \text{attack} = l_2)$

     $\mathcal{B}^{\tau_\infty}(\boldsymbol{x}'_\infty) \leftarrow \text{Box}(\boldsymbol{x}'_\infty, \tau_\infty)$       // Get box with midpoint $\boldsymbol{x}'_\infty, \boldsymbol{x}'_2$ and radius $\tau_\infty, \tau_2$

     $\mathcal{B}^{\tau_2}(\boldsymbol{x}'_2) \leftarrow \text{Box}(\boldsymbol{x}'_2, \tau_2)$

     $\boldsymbol{u}_{y^\Delta_\infty} \leftarrow \text{get\_upper\_bound}(f_\theta, \mathcal{B}^{\tau_\infty}(\boldsymbol{x}'_\infty))$       // Get upper bound $\boldsymbol{u}_{y^\Delta_\infty}, \boldsymbol{u}_{y^\Delta_2}$ on logit differences

     $\boldsymbol{u}_{y^\Delta_2} \leftarrow \text{get\_upper\_bound}(f_\theta, \mathcal{B}^{\tau_2}(\boldsymbol{x}'_2))$       // based on IBP

     $\text{loss}_{l_\infty} \leftarrow \mathcal{L}_{\text{CE}}(\boldsymbol{u}_{y^\Delta_\infty}, t)$

     $\text{loss}_{l_2} \leftarrow \mathcal{L}_{\text{CE}}(\boldsymbol{u}_{y^\Delta_2}, t)$

     $\text{loss}_{\ell_1} \leftarrow \ell_1 \cdot \text{get\_}\ell_1\text{\_norm}(f_\theta)$

     $\text{loss}_{tot} \leftarrow \text{loss}_{l_\infty} + \text{loss}_{l_2} + \text{loss}_{\ell_1}$

     $\theta \leftarrow \theta - \eta \cdot \nabla_\theta \text{loss}_{tot}$       // Update model parameters $\theta$

   **end for**

---

**Algorithm 5** CURE-Scratch/Finetune Training Epoch

---

**Require:** Neural network $f_\theta$, training set $(\boldsymbol{X}, \boldsymbol{T})$, perturbation radius $\epsilon_2$ and $\epsilon_\infty$, subselection ratio $\lambda_\infty$ and $\lambda_2$, learning rate $\eta$, $\ell_1$ regularization weight $\ell_1$, KL loss balance factor $\eta$, mode $\in [\text{Scratch, Finetune}]$

   **for** $(\boldsymbol{x}, t) = (\boldsymbol{x}_0, t_0) \ldots (\boldsymbol{x}_b, t_b)$ **do**       // Sample batches $\sim (\boldsymbol{X}, \boldsymbol{T})$

     $(\boldsymbol{x}'_\infty, \tau_\infty) \leftarrow \text{get\_propagation\_region}(\text{attack} = l_\infty)$ // Refer to Algorithm 1

     $(\boldsymbol{x}'_2, \tau_2) \leftarrow \text{get\_propagation\_region}(\text{attack} = l_2)$

     $\mathcal{B}^{\tau_\infty}(\boldsymbol{x}'_\infty) \leftarrow \text{Box}(\boldsymbol{x}'_\infty, \tau_\infty)$       // Get box with midpoint $\boldsymbol{x}'_\infty, \boldsymbol{x}'_2$ and radius $\tau_\infty, \tau_2$

     $\mathcal{B}^{\tau_2}(\boldsymbol{x}'_2) \leftarrow \text{Box}(\boldsymbol{x}'_2, \tau_2)$

     $\boldsymbol{u}_{y^\Delta_\infty} \leftarrow \text{get\_upper\_bound}(f_\theta, \mathcal{B}^{\tau_\infty}(\boldsymbol{x}'_\infty))$       // Get upper bound $\boldsymbol{u}_{y^\Delta_\infty}, \boldsymbol{u}_{y^\Delta_2}$ on logit differences

     $\boldsymbol{u}_{y^\Delta_2} \leftarrow \text{get\_upper\_bound}(f_\theta, \mathcal{B}^{\tau_2}(\boldsymbol{x}'_2))$       // based on IBP

     $\text{loss}_{l_\infty} \leftarrow \mathcal{L}_{\text{CE}}(\boldsymbol{u}_{y^\Delta_\infty}, t)$

     $\text{loss}_{l_2} \leftarrow \mathcal{L}_{\text{CE}}(\boldsymbol{u}_{y^\Delta_2}, t)$

     $\text{loss}_{Max} \leftarrow max(\text{loss}_{l_\infty}, \text{loss}_{l_2})$       // We select the largest $l_{p \in [2, \infty]}$ loss for each sample

     $\text{loss}_{\ell_1} \leftarrow \ell_1 \cdot \text{get\_}\ell_1\text{\_norm}(f_\theta)$

     find correctly certified $l_q$ subset $\gamma$ using Definition 4.3

     $\text{loss}_{KL} \leftarrow KL(d_q[\gamma] \| d_r[\gamma])$       // Eq. 5

     $\text{loss}_{tot} \leftarrow \text{loss}_{Max} + \eta \cdot \text{loss}_{KL} + \text{loss}_{\ell_1}$

     $\theta \leftarrow \theta - \eta \cdot \nabla_\theta \text{loss}_{tot}$       // Update model parameters $\theta$

   **end for**

---

---

**Algorithm 6** GP: Connect CT with NT

---

1: **Input**: model $f_\theta$, input images with distribution $\mathcal{D}$, training rounds $R$, $\beta$, natural training **NT** and certified training **CT** algorithms, perturbation radius $\epsilon_\infty$ and $\epsilon_2$, subselection ratio $\lambda_\infty$ and $\lambda_2$, learning rate $\eta$, $\ell_1$ regularization weight $\ell_1$.

2:

3: **for** $r = 1, 2, ..., R$ **do**

4:     $f_n \leftarrow \mathbf{NT}(f_\theta^{(r)}, \mathcal{D})$

5:     $f_c \leftarrow \mathbf{CT}(f_\theta^{(r)}, \epsilon_\infty, \epsilon_2, \lambda_\infty, \lambda_2, \eta, \ell_1, \mathcal{D})$            // Can be single-norm or any CURE training

6:     compute $g_n \leftarrow f_n - f_\theta^{(r)}$, $g_c \leftarrow f_c - f_\theta^{(r)}$

7:     compute $g_p$ using Eq. 10

8:     update $f_\theta^{(r+1)}$ using Eq. 11 with $\beta$ and $g_c$

9: **end for**

10: **Output**: model $f_\theta$.

---

