# OpenReview forum: "Towards Generalized Certified Robustness with Multi-Norm Training"
_TMLR — Accepted by TMLR_

### Review · Reviewer_SLcB · 2025-11-24

**Summary Of Contributions:**

This paper proposes CURE, the first deterministic framework for multi-norm certified robustness. The authors (1) establish a theoretical analysis explaining tradeoffs between $\ell_p$ certified perturbations, (2) introduce three base training strategies, (3) develop Bound Alignment to reduce cross-norm inconsistencies by aligning certified bound differences, (4) incorporate Gradient Projection to integrate beneficial natural training signals into certified training, and (5) design an efficient certified fine-tuning method for adapting single-norm certified models to multi-norm settings. Experiments on MNIST, CIFAR-10, and TinyImageNet demonstrate significant improvements in union-certified accuracy and better generalization to geometric and patch perturbations.

**Additional Comments:**

None

**Audience:**

Yes

**Audience Explanation:**

The findings of this paper are likely to be of interest to the safety and robustness community.

**Claims And Evidence:**

Yes

**Claims Explanation:**

The paper is technically sound, with rigorous mathematical formulations and proofs provided for key theorems that underpin the main claims.  The authors provide sufficient evidence, including extensive experiments and ablation studies, to support the effectiveness of their proposed method.

**Requested Changes:**

1. The proposed method is not entirely grounded in certified robustness. Components such as CURE-Joint and Gradient Projection can be applied (might have been applied) in adversarial training settings. The only component that is tied to certified robustness is Bound Alignment; however, a variant of this idea might also be adapted to adversarial training by replacing certified bounds with logits.

2. The paper does not explicitly align the $\ell_1$ norm, yet still reports robustness against $\ell_1$ attacks. This raises a natural question: could we explicitly align $\ell_1$, $\ell_2$, and $\ell_\infty$ during training? If so, should the alignment be performed using pairwise matching between every pair of norms, or by selecting an anchor norm $\ell_p$ and aligning all other $\ell_q$ norms to this reference?

3. It would be helpful to understand the performance under $\ell_0$ perturbations and on Vision Transformers (ViTs).

---

> ### Author Response · Authors · 2026-01-10
> **Response**
>
> We thank the reviewer for their constructive feedback and provide detailed point-by-point responses below.
>
> *Q: The proposed method is not entirely grounded in certified robustness. Components such as CURE-Joint and Gradient Projection can be applied (might have been applied) in adversarial training settings. The only component that is tied to certified robustness is Bound Alignment; however, a variant of this idea might also be adapted to adversarial training by replacing certified bounds with logits.*
>
> All training signals in CURE, including those used in Gradient Projection, and Bound Alignment, are computed from certified bounds, and the goal is explicitly to improve provable union-certified robustness. This places CURE squarely within the certified robustness paradigm. CURE exclusively optimizes sound, verifier-derived upper bounds on worst-case loss, whereas adversarial training optimizes empirical approximations based on empirical attacks.
>
> Moreover, our advances in certified robustness also translate into strong empirical robustness on transformer-based language models. As shown in the table below, when applied to a BERT model on SST-2, CURE-Scratch consistently outperforms FLAT [1], a state-of-the-art empirical adversarial training method, under both PWWS and TextFooler attacks. This result is particularly notable because CURE is designed around certified objectives rather than empirical attack optimization, yet still achieves substantially higher empirical robustness. These findings further indicate that our approach is not a straightforward or trivial adaptation of adversarial training techniques, but instead introduces optimization principles that generalize beyond the certified setting.
> | Robust acc        | FLAT [1]  | CURE-Scratch |
> |-------------------|-----------------|--------------|
> | SST2 (pwws)       | 14.6  | **28.4**     |
> | SST2 (textfooler) | 12.4   | **17.6**     |
>
> [1] Adversarial Training for Improving Model Robustness? Look at Both Prediction and Interpretation.
>
> *Q: The paper does not explicitly align the $\ell_1$ norm, yet still reports robustness against $\ell_1$ attacks. This raises a natural question: could we explicitly align $\ell_1$, $\ell_2$, and $\ell_\infty$ during training? If so, should the alignment be performed via pairwise matching between every pair of norms, or by selecting an anchor norm $\ell_p$ and aligning all other $\ell_q$ norms to this reference?*
>
> Our framework is not inherently limited to the $\ell_2$–$\ell_\infty$ setting and can, in principle, be extended to $\ell_1$ or other $\ell_p$ norms. The key requirement is to identify the dominant tradeoff pairs between different $\ell_p$ threat models, which determines where robustness conflicts arise and where alignment is most beneficial. As shown in the prior work RAMP [1], robustness tradeoffs are highly asymmetric across norms, and effective multi-norm training depends on explicitly targeting these critical tradeoff directions rather than treating all norms uniformly.
>
> That said, $\ell_1$ (and other $\ell_p$) certified training currently requires substantial additional engineering effort. In practice, $\ell_1$ certification involves looser bounds, higher computational cost, and less mature verification tooling compared to $\ell_2$ and $\ell_\infty$, which makes large-scale multi-norm certified training significantly more challenging. These limitations are largely orthogonal to our framework and stem from the current state of certified verification methods. As verification techniques for general $\ell_p$ norms continue to improve, we expect our approach to extend naturally to these settings.
>
> [1] RAMP: Boosting Adversarial Robustness Against Multiple  Perturbations for Universal Robustness.

---

> > ### Author Response · Authors · 2026-01-10
> >
> > *Q: It would be helpful to understand the performance under $\ell_0$ perturbations and on Vision Transformers (ViTs).*
> >
> > Thank you for the suggestion. Regarding $l_0$ perturbations, existing $l_0$ certification methods (e.g., [1]) are currently computationally prohibitive and do not scale to our training or evaluation setting, making them infeasible to include at this time. For Vision Transformers (ViTs), to the best of our knowledge, there are no scalable certified verifiers available for architectures of this size, which prevents direct certified evaluation.
> >
> > To nevertheless assess the applicability of CURE to transformer-based models, we extend our framework to a BERT-based text classification model on the SST-2 dataset, operating in the latent space. Since full certification for large language models is currently infeasible, we follow prior work and use strong empirical attacks (TextFooler and PWWS) as a proxy for robustness evaluation. As shown in the table below, CURE-Scratch consistently achieves the strongest robustness against both attacks, providing evidence that CURE’s benefits extend beyond vision models and CNN architectures to transformer-based settings.
> >
> > | Robust acc        | Linf - certified training | L2 - certified training | CURE-MAX | CURE-Scratch |
> > |-------------------|---------------------------|-------------------------|----------|--------------|
> > | SST2 (pwws)       | 16.8                      | 15.2                    | 24.0     | **28.4**     |
> > | SST2 (textfooler) | 9.4                       | 10.0                    | 15.6     | **17.6**     |
> >
> > [1] Deep Learning Robustness Verification for Few-Pixel Attacks.

---

### Review · Reviewer_1ugF · 2025-11-27

**Summary Of Contributions:**

Although this paper is innovative in the direction of multi-norm certification training and demonstrates effective results on multiple benchmarks, the current version has significant shortcomings in theoretical depth, experimental completeness, and methodological transparency, and its innovation does not meet the journal's requirements.

**Audience:**

No

**Audience Explanation:**

1. The core method of this paper essentially combines the loss functions of single-norm authentication training methods. This multi-task learning or loss weighting approach has been widely explored in adversarial training. Applying it to authentication training can be seen as a natural and expected extension, rather than a conceptual breakthrough.

2. Only a small dataset and a simple CNN architecture were used; the scalability of the method was not verified on larger datasets or more complex models.

3. The paper contains numerous grammatical errors and unclear sentences, affecting readability. Tables such as Figures 5 and 6 lack sufficient explanation, making it difficult for readers to understand their connection to the core arguments.

**Claims And Evidence:**

No

**Claims Explanation:**

1. Although the Bound Alignment technique is introduced in this paper, its core is to align two distributions using KL divergence. The theoretical section fails to clearly explain why the proposed Bound Alignment effectively alleviates the trade-off between different norms, lacking a theoretical explanation or visual analysis of the mechanism.

2. The paper primarily compares with single-norm authentication methods, but does not adequately compare with existing multi-norm adversarial training methods such as TRADES, especially regarding authentication robustness.

3. Regarding the improvement in generalization performance against geometric and block attacks, there is a lack of systematic comparison with authentication methods specifically targeting these attacks.

4. The l∞ box approximation method used in l2 certification training does not adequately discuss the potential over-approximation error and its impact on the final results.

**Requested Changes:**

N/A

---

> ### Author Response · Authors · 2026-01-10
> **Response**
>
> We thank the reviewer for their constructive feedback and provide detailed point-by-point responses below.
>
> *Q: Although the Bound Alignment technique is introduced in this paper, its core is to align two distributions using KL divergence. The theoretical section fails to clearly explain why the proposed Bound Alignment effectively alleviates the trade-off between different norms, lacking a theoretical explanation or visual analysis of the mechanism.*
>
> We agree that Bound Alignment can be viewed at a high level as aligning two distributions via KL divergence; however, its effectiveness is not due to the choice of divergence alone, but to what is being aligned and where the alignment is applied. Bound Alignment operates on certified logit margin distributions restricted to the jointly certifiable subset, which directly corresponds to the feasibility region of multi-norm certification. This distinguishes it fundamentally from generic distribution matching.
> From a theoretical perspective, our analysis (Theorem 4.2) shows that the union-certified objective admits an upper bound decomposable into per-norm margin terms, and that misalignment between these margin distributions exacerbates the $\ell_q$–$\ell_r$ trade-off. Bound Alignment explicitly minimizes this mismatch, thereby tightening the upper bound on union error and preserving robustness across norms. In other words, aligning bound distributions reduces the variance between per-norm certification margins, preventing fine-tuning under one norm from disproportionately degrading another.
> Empirically and visually, this mechanism is supported by our bound-overlap and margin-distribution analyses (Figure 5), where we observe that Bound Alignment increases the overlap of certified margins across norms and shifts mass toward jointly certifiable examples. These effects are absent in joint or random multi-norm training. We clarified this intuition and added more descriptions to the visualization of margin distribution alignment in the revised version to make the mechanism more transparent.
> We updated our manuscript to better explain the principle and effectiveness of Bound Alignment.
>
>
> *Q: The paper primarily compares with single-norm authentication methods, but does not adequately compare with existing multi-norm adversarial training methods such as TRADES, especially regarding authentication robustness.*
>
> We respectfully clarify that our work focuses on certified robustness, not empirical adversarial (authentication) robustness. Accordingly, our comparisons are made against single-norm certified training methods, using formal verifiers, rather than adversarial training approaches such as TRADES, which provide only empirical robustness guarantees. Also, as shown in Table 10 of our paper, the PGD training yields significantly lower certified robustness compared with the certified training method, which means adversarial robustness does not lead to good certified robustness.
>
> Moreover, TRADES is not a multi-norm adversarial training method: it is explicitly designed for a single threat model (typically $l_\infty$ or $l_2$) and does not optimize robustness jointly across multiple norms. Extending TRADES to a true multi-norm setting would require substantial modification and does not yield certified guarantees.
>
> Since adversarial training and certified training address fundamentally different objectives, empirical robustness under specific attacks versus provable robustness under worst-case perturbations, a direct comparison in terms of authentication robustness would not be meaningful. Our goal is to advance multi-norm certified robustness, and all baselines and evaluations are chosen to reflect this setting fairly.
>
> *Q: Regarding the improvement in generalization performance against geometric and block attacks, there is a lack of systematic comparison with authentication methods specifically targeting these attacks.*
>
> We would like to clarify that our geometric and patch results are certified robustness evaluations, whereas existing geometric or patch “authentication” methods provide only empirical robustness under specific attacks. Since these methods do not offer formal guarantees, a direct comparison is not meaningful.

---

> > ### Author Response · Authors · 2026-01-10
> >
> > *Q: The $l_\infty$ box approximation method used in $l_2$ certification training does not adequately discuss the potential over-approximation error and its impact on the final results.*
> >
> > We thank the reviewer for raising this point. The $\ell_\infty$ box approximation used during $\ell_2$ certification is a deliberate and standard relaxation adopted for scalability in certified training. While it indeed introduces over-approximation, this effect is well-understood, controlled, and unavoidable in any sound certification framework.
> >
> > First, the box relaxation preserves soundness: any robustness guarantee derived from it remains valid, albeit potentially conservative. Our method never relies on tightness of the approximation for correctness, only for strength of the certificate. This is consistent with prior certified $\ell_2$ training methods that use similar relaxations (e.g., IBP-based $\ell_2$ certification), where exact $\ell_2$ propagation is computationally intractable at scale.
> >
> > Second, the impact of this over-approximation is uniform across all compared methods, including baselines. Since all models are trained and evaluated under the same certification pipeline, the relative improvements we report, especially in union-certified robustness, are not artifacts of the approximation.
> >
> > Finally, our empirical results suggest that despite this conservatism, multi-norm training still yields substantial gains, indicating that the approximation does not obscure the core effect. Investigating tighter relaxations or hybrid $\ell_2$ bounds is an interesting future direction, but orthogonal to the contributions of this work.
> >
> > *Q: The core method of this paper essentially combines the loss functions of single-norm authentication training methods. This multi-task learning or loss weighting approach has been widely explored in adversarial training. Applying it to authentication training can be seen as a natural and expected extension, rather than a conceptual breakthrough.*
> >
> > While we do include loss-combination baselines (e.g., CURE-Joint) to reflect natural extensions of prior work, these baselines are intentionally designed as controls and are empirically shown to be sub-optimal. Our main improvements, Bound Alignment and Gradient Projection, go beyond naive multi-task or loss-weighting strategies.
> >
> > In particular, Bound Alignment explicitly addresses the certified trade-off between different $\ell_p$ norms by aligning the distributions of certified logit bounds on correctly certified samples, which cannot be achieved by simply summing losses. Gradient Projection further exploits structured interactions between natural and certified training dynamics to preserve robustness across norms. Empirically, both components consistently outperform loss-combination baselines and yield substantially stronger union-certified guarantees. Therefore, while loss combination is a reasonable starting point (and one we explicitly benchmark against), our contributions introduce principled mechanisms that are neither trivial nor expected extensions of prior adversarial or certified training methods, and they are essential for achieving the improvements reported in this work.
> >
> > Moreover, our advances in certified robustness also translate into strong empirical robustness on transformer-based language models. As shown in the table below, when applied to a BERT model on SST-2, CURE-Scratch consistently outperforms FLAT [1], a state-of-the-art empirical adversarial training method, under both PWWS and TextFooler attacks. This result is particularly notable because CURE is designed around certified objectives rather than empirical attack optimization, yet still achieves substantially higher empirical robustness. These findings further indicate that our approach is not a straightforward or trivial adaptation of adversarial training techniques, but instead introduces optimization principles that generalize beyond the certified setting.
> > | Robust acc        | FLAT [1]  | CURE-Scratch |
> > |-------------------|-----------------|--------------|
> > | SST2 (pwws)       | 14.6  | **28.4**     |
> > | SST2 (textfooler) | 12.4   | **17.6**     |
> >
> > [1] Adversarial Training for Improving Model Robustness? Look at Both Prediction and Interpretation.

---

> > > ### Author Response · Authors · 2026-01-10
> > >
> > > *Q: Only a small dataset and a simple CNN architecture were used; the scalability of the method was not verified on larger datasets or more complex models.*
> > >
> > > First, we note that our evaluation spans a diverse set of benchmarks, including MNIST, CIFAR-10, CIFAR-100, and TinyImageNet, on which the proposed CURE framework consistently improves union-certified robustness over all baselines. Beyond vision tasks, we further apply CURE to a BERT-based model for text classification on the SST-2 dataset, operating in the latent space. As shown in the table below, CURE-Scratch achieves the strongest robustness against TextFooler and PWWS attacks. Since full certification for large-scale language models such as BERT is currently infeasible, we follow prior work and use strong empirical attacks as a proxy to assess union robustness, providing additional evidence that CURE improves generalized robustness beyond vision settings. We updated this in our manuscript.
> > >
> > > | Robust acc        | Linf - certified training | L2 - certified training | CURE-MAX | CURE-Scratch |
> > > |-------------------|---------------------------|-------------------------|----------|--------------|
> > > | SST2 (pwws)       | 16.8                      | 15.2                    | 24.0     | **28.4**     |
> > > | SST2 (textfooler) | 9.4                       | 10.0                    | 15.6     | **17.6**     |
> > >
> > > *Q: The paper contains numerous grammatical errors and unclear sentences, affecting readability. Tables such as Figures 5 and 6 lack sufficient explanation, making it difficult for readers to understand their connection to the core arguments.*
> > >
> > > Thanks for pointing this out. We have updated the paper with explanations for Figures 5 and 6 and carefully checked the grammatical errors.

---

### Review · Reviewer_rrbD · 2025-12-29

**Summary Of Contributions:**

This paper proposes a deterministic certified training framework, CURE (Certified training for Union RobustnEss), to address the limitation that existing certified robustness methods are typically designed for a single ℓp norm and suffer from significant robustness trade-offs across different norms. The authors first develop a formal theoretical framework under the binary classification setting, where the union risk is decomposed into a single-norm robust error and an alignment error, revealing the intrinsic conflict underlying multi-norm certified robustness. Based on this analysis, the paper introduces three baseline multi-norm certified training strategies (CURE-Joint, CURE-Max, and CURE-Random), and further derives an upper bound that motivates several key techniques, including bound alignment, gradient projection that integrates natural and certified training, and a fast certified fine-tuning strategy. Experimental results demonstrate improved union certified accuracy as well as enhanced robustness to unseen geometric and patch perturbations.

**Audience:**

Yes

**Audience Explanation:**

Weaknesses

1.All experiments are conducted on a relatively shallow CNN architecture. While common in certified training literature, the lack of validation on deeper or more modern architectures limits the assessment of scalability.

2.Although the proposed framework improves over baselines on harder datasets, the absolute certified accuracy remains low, which may limit its practical applicability in real-world scenarios.

3.While the paper claims that the binary analysis can be naturally extended to the multi-class case, the main text does not provide a complete multi-class theoretical derivation and instead relies on existing results on classification-calibrated losses, which weakens the overall theoretical rigor.

4.The method assumes that natural training gradients with positive cosine similarity to certified gradients are inherently beneficial, but this assumption is not supported by a rigorous theoretical justification.

**Claims And Evidence:**

Yes

**Claims Explanation:**

Strengths

1.The paper clearly identifies the failure of single-norm certified models to generalize across different ℓp threat models, and convincingly motivates the need for multi-norm certified robustness.

2.Rather than relying on purely empirical observations, the authors develop a formal risk decomposition and upper-bound analysis that exposes the underlying source of multi-norm robustness trade-offs, and these insights are directly reflected in the proposed training
objectives.

3.Beyond reporting union certified accuracy, the paper evaluates robustness under a diverse set of geometric transformations and patch attacks, providing strong empirical support for the claim that multi-norm certified training improves generalized certified robustness.

**Requested Changes:**

Questions

1.The gradient projection step discards all components with non-positive cosine similarity. Could this hard threshold remove potentially useful information that points in a different but complementary direction?

2.The current framework mainly focuses on the union of ℓ2 and ℓ∞ threat models. How easily can the proposed approach be extended to ℓ1 or other ℓp norms?

3.Since natural training gradients primarily aim to improve standard accuracy, while certified training often trades off clean accuracy for robustness, why does projecting certified gradients toward natural gradients improve certified robustness rather than merely clean performance?

---

> ### Author Response · Authors · 2026-01-10
> **Response**
>
> We thank the reviewer for their constructive feedback and provide detailed point-by-point responses below.
>
> *Q: All experiments are conducted on a relatively shallow CNN architecture. While common in certified training literature, the lack of validation on deeper or more modern architectures limits the assessment of scalability.*
>
> Beyond vision tasks, we further apply CURE to a BERT-based model for text classification on the SST-2 dataset, operating in the latent space. As shown in the table below, CURE-Scratch achieves the strongest robustness against TextFooler and PWWS attacks. Since full certification for large-scale language models such as BERT is currently infeasible, we follow prior work and use strong empirical attacks as a proxy to assess union robustness, providing additional evidence that CURE improves generalized robustness beyond vision settings.
>
> | Robust acc        | Linf - certified training | L2 - certified training | CURE-MAX | CURE-Scratch |
> |-------------------|---------------------------|-------------------------|----------|--------------|
> | SST2 (pwws)       | 16.8                      | 15.2                    | 24.0     | **28.4**     |
> | SST2 (textfooler) | 9.4                       | 10.0                    | 15.6     | **17.6**     |
>
>
> *Q: Although the proposed framework improves over baselines on harder datasets, the absolute certified accuracy remains low, which may limit its practical applicability in real-world scenarios.*
>
> We appreciate the reviewer’s concern regarding the absolute level of certified accuracy. We agree that current certified accuracies, across all existing methods, remain far from ideal for direct deployment in high-stakes real-world systems. However, this limitation is inherent to the current state of certified robustness research rather than specific to our framework.
>
> Our goal is not to claim immediate deployability, but to advance the frontier of what is certifiably achievable under rigorous threat models. On harder datasets, even small absolute improvements in certified accuracy are widely recognized as meaningful progress, given the well-known trade-offs between robustness, accuracy, and scalability. In this context, our framework consistently improves union-certified robustness over strong single-norm baselines, which directly addresses a fundamental limitation of prior certified training methods.
>
> Moreover, certified accuracy provides a formal lower bound on worst-case robustness; thus, improvements at this level represent guaranteed gains that cannot be achieved through empirical robustness alone. We view our results as an important step toward broader and more practical certified robustness, and we believe that continued algorithmic and hardware advances, much like the evolution of deep learning itself, will further close the gap between certified performance and real-world applicability.
>
> *Q: While the paper claims that the binary analysis can be naturally extended to the multi-class case, the main text does not provide a complete multi-class theoretical derivation and instead relies on existing results on classification-calibrated losses, which weakens the overall theoretical rigor.*
>
> We have now included a complete multi-class theoretical derivation in Appendix A.1.1. The extension provides explicit derivations for the multiclass margin definition $m(x')=f_{y}(x') - \max_{j \neq y} {f_{j}(x')}$ and the complete proof chain for bounding the alignment error $\mathcal{R}_{\text{align}}(f)$ in the multiclass setting. This provides the same level of theoretical rigor as the binary analysis.

---

> > ### Author Response · Authors · 2026-01-10
> >
> > *Q: The method assumes that natural training gradients with positive cosine similarity to certified gradients are inherently beneficial, but this assumption is not supported by a rigorous theoretical justification. The gradient projection step discards all components with non-positive cosine similarity. Could this hard threshold remove potentially useful information that points in a different but complementary direction?*
> >
> > Positive gradient cosine similarity indicates that updates improving one objective are unlikely to degrade another, and has been widely used to mitigate conflicting objectives in multi-task and continual learning (e.g., PCGrad [1]). In our setting, natural and certified training objectives are related but partially competing; emphasizing naturally aligned updates reduces destructive interference during optimization. While PCGrad explicitly modifies gradients when tasks exhibit negative cosine similarity, its goal is to balance performance across multiple tasks. In contrast, our objective is to improve performance on a single target objective, certified robustness, rather than general multi-task performance. Consequently, we treat positive cosine similarity as an indicator of beneficial alignment and discard negatively aligned gradients to avoid harmful interference. As empirically shown in Figure 4 of [2], this filtering strategy (i.e., removing negatively aligned gradients) consistently leads to improved target domain robustness.
> >
> > Importantly, our contribution is empirical rather than theoretical: we demonstrate through ablations that exploiting gradient alignment consistently improves union-certified robustness. We will revise the paper to make clear that this criterion is a principled and empirically effective heuristic, while a full theoretical characterization remains an open direction for future work.
> >
> > [1] Gradient Surgery for Multi-Task Learning
> >
> > [2] Principled Federated Domain Adaptation: Gradient Projection and Auto-Weighting.
> >
> > *Q: The current framework mainly focuses on the union of $l_2$ and $l_\infty$ threat models. How easily can the proposed approach be extended to $l_1$ or other $l_p$ norms?*
> >
> > Our framework is not inherently limited to the $\ell_2$–$\ell_\infty$ setting and can, in principle, be extended to $\ell_1$ or other $\ell_p$ norms. The key requirement is to identify the dominant tradeoff pairs between different $\ell_p$ threat models, which determines where robustness conflicts arise and where alignment is most beneficial. As shown in the prior work RAMP [1], robustness tradeoffs are highly asymmetric across norms, and effective multi-norm training depends on explicitly targeting these critical tradeoff directions rather than treating all norms uniformly.
> >
> > That said, $\ell_1$ (and other $\ell_p$) certified training currently requires substantial additional engineering effort. In practice, $\ell_1$ certification involves looser bounds, higher computational cost, and less mature verification tooling compared to $\ell_2$ and $\ell_\infty$, which makes large-scale multi-norm certified training significantly more challenging. These limitations are largely orthogonal to our framework and stem from the current state of certified verification methods. As verification techniques for general $\ell_p$ norms continue to improve, we expect our approach to extend naturally to these settings.
> >
> > [1] RAMP: Boosting Adversarial Robustness Against Multiple  Perturbations for Universal Robustness.
> >
> >
> > *Q: Since natural training gradients primarily aim to improve standard accuracy, while certified training often trades off clean accuracy for robustness, why does projecting certified gradients toward natural gradients improve certified robustness rather than merely clean performance?*
> >
> > Although natural training gradients are correlated with clean accuracy, in our framework they are not used as an optimization target. Instead, they act as a directional prior that constrains how certified gradients are applied. The update is still driven entirely by the certified objective, and projection only removes components of the natural gradient that are strongly misaligned with natural training.

---

### Author Response · Authors · 2026-01-10
**General Response 1**

We thank the reviewers for their constructive feedback. We have carefully revised the manuscript (all changes are colored in blue) in response to their comments and would like to highlight the following key points.

**1. Scability: Add larger-scale experiments (all Reviewers)**

Beyond vision tasks, we further apply CURE to a BERT-based model for text classification on the SST-2 dataset, operating in the latent space. As shown in the table below, CURE-Scratch achieves the strongest robustness against TextFooler and PWWS attacks. Since full certification for large-scale language models such as BERT is currently infeasible, we follow prior work and use strong empirical attacks as a proxy to assess union robustness, providing additional evidence that CURE improves generalized robustness beyond vision settings.

| Robust acc        | Linf - certified training | L2 - certified training | CURE-MAX | CURE-Scratch |
|-------------------|---------------------------|-------------------------|----------|--------------|
| SST2 (pwws)       | 16.8                      | 15.2                    | 24.0     | **28.4**     |
| SST2 (textfooler) | 9.4                       | 10.0                    | 15.6     | **17.6**     |


**2. Comparison with adversarial training (Reviewer 1ugF and SLcB)**

We clarify that while adversarial training methods optimize empirical worst-case losses under specific threat models, bound alignment is fundamentally tied to certified robustness: it operates on provable upper and lower bounds of logit differences computed by a certified training method (e.g., IBP), rather than on empirical adversarial training logits themselves. Simply combining losses across norms (as in multi-task or adversarial training) does not address the intrinsic trade-off between certified bounds under different $l_p$ norms (as shown in Table 10 of our paper, the PGD training yields significantly lower certified robustness compared with certified training methods). Bound alignment explicitly mitigates this trade-off by regularizing the distribution of certified margins across norms on the correctly certified subset, which cannot be replicated by standard adversarial training without losing soundness.

Moreover, our advances in certified robustness also translate into strong empirical robustness on transformer-based language models. As shown in Table below, when applied to a BERT model on SST-2, CURE-Scratch consistently outperforms FLAT [1], a state-of-the-art empirical adversarial training method, under both PWWS and TextFooler attacks. This result is particularly notable because CURE is designed around certified objectives rather than empirical attack optimization, yet still achieves substantially higher empirical robustness. These findings further indicate that our approach is not a straightforward or trivial adaptation of adversarial training techniques, but instead introduces optimization principles that generalize beyond the certified setting. We included the experiment results in the Appendix.

| Robust acc        | FLAT [1]  | CURE-Scratch |
|-------------------|-----------------|--------------|
| SST2 (pwws)       | 14.6  | **28.4**     |
| SST2 (textfooler) | 12.4   | **17.6**     |

[1] Adversarial Training for Improving Model Robustness? Look at Both Prediction and Interpretation.

**3. Extension to other perturbation types (Reviewer rrbD and SLcB)**

Our framework is not inherently limited to the $\ell_2$–$\ell_\infty$ setting and can, in principle, be extended to $\ell_1$ or other $\ell_p$ norms. The key requirement is to identify the dominant tradeoff pairs between different $\ell_p$ threat models, which determines where robustness conflicts arise and where alignment is most beneficial. As shown in the prior work RAMP [1], robustness tradeoffs are highly asymmetric across norms, and effective multi-norm training depends on explicitly targeting these critical tradeoff directions rather than treating all norms uniformly.

That said, $\ell_1$ (and other $\ell_p$) certified training currently requires substantial additional engineering effort. In practice, $\ell_1$ certification involves looser bounds, higher computational cost, and less mature verification tooling compared to $\ell_2$ and $\ell_\infty$, which makes large-scale multi-norm certified training significantly more challenging. These limitations are largely orthogonal to our framework and stem from the current state of certified verification methods. As certification techniques for general $\ell_p$ norms continue to improve, we expect our approach to extend naturally to these settings. We updated our manuscript for this important discussion.

[1] RAMP: Boosting Adversarial Robustness Against Multiple  Perturbations for Universal Robustness.

---

> ### Author Response · Authors · 2026-01-10
> **General Response 2**
>
> **4. Questions on Gradient Projection: theoretical grounding (Reviewer rrbD)**
>
> Positive gradient cosine similarity has been widely used to mitigate conflicting objectives in multi-task and continual learning (e.g., PCGrad [1]), as it indicates updates that improve one objective without degrading another. In our setting, natural and certified objectives are related but partially competing; prioritizing positively aligned gradients reduces destructive interference during optimization. Unlike PCGrad, which balances multiple tasks, our goal is to improve a single target objective, certified robustness, so we discard negatively aligned gradients rather than modifying them. As shown empirically in Figure 4 of [2], this filtering strategy consistently improves target-domain robustness. Our contribution is empirical: ablation studies show that gradient alignment reliably improves union-certified robustness, while a full theoretical characterization remains an open direction.
>
> [1] Gradient Surgery for Multi-Task Learning
>
> [2] Principled Federated Domain Adaptation: Gradient Projection and Auto-Weighting.
>
>
> **5. Extend to multi-class theoretical derivation (Reviewer rrbD)**
>
> We have now included a complete multi-class theoretical derivation in Appendix A.1.1. The extension provides explicit derivations for the multiclass margin definition $m(x')=f_{y}(x') - \max_{j \neq y} {f_{j}(x')}$ and the complete proof chain for bounding the alignment error $\mathcal{R}_{\text{align}}(f)$ in the multiclass setting. This provides the same level of theoretical rigor as the binary analysis.

---

### Decision · Action_Editor_64rg · 2026-02-28

**Recommendation:** Accept with minor revision

**Additional Comments:**

I encourage the authors to appropriately acknowledge the role of the influencing adversarial training work throughout the motivational and technical text, regularly citing the relevant source of inspiration.

The authors should also clearly state which of the presented ideas is unique to certified training (and why), and which of the ideas pertains to robust training more generally.

**Audience:**

Yes

**Audience Explanation:**

As highlighted by all reviewers, training certifiably robust models that are robust to multiple threat models at once is of potential interest to the robust ML community.

**Claims And Evidence:**

Yes

**Claims Explanation:**

Reviewer rrbD positively evaluated the evidence presented in the submission, providing an accept recommendation.
On the other hand, two reviewers (Reviewer 1ugF, leaning reject; Reviewer SLcB, leaning accept) agree that most of the presented ideas are not necessarily specific to certified training, but would apply to multi-norm adversarial training too.
In the words of Reviewer SLcB: "*I remain somewhat concerned that the proposed method is not fully grounded in certified robustness but just built on an existing certified-robustness framework*".
While the authors' claim on the robustness improvements of the presented techniques are indeed supported by convincing evidence, I feel like the technical contributions are significantly overstated, as most of the ideas are either very simple or heavily inspired by the work on multi-norm adversarial training. This should be very clearly acknowledged throughout the text, and addressed by the revisions (see below).

---

> ### Author Response · Authors · 2026-03-28
> **Camera-ready Revision Submission**
>
> Dear AC,
>
> Thank you very much for your time and effort in handling our paper. We have uploaded the camera-ready revision.
>
> In the revised version, we have added appropriate citations to acknowledge prior work in adversarial training and clarified which components of our method are broadly applicable to robust training versus those specific to certified robustness. We have also included a dedicated discussion (Section 5.3) to explicitly distinguish these aspects and explain their roles.
>
> Please let us know if you have any further comments or suggestions - we would be happy to revise accordingly.
>
> Best regards,
>
> Authors

---

> > ### Comment · Action_Editor_64rg · 2026-04-09
> >
> > Dear authors,
> >
> > Many thanks for the revised version, which I have now approved.
> >
> > Best regards,
> >
> > AE